# VERIFY: A Novel Multi-Domain Dataset Grounding LTL in Contextual Natural Language via Provable Intermediate Logic

**Paapa Kwesi Quansah,    Pablo Rivas Perea,    Ernest Bonnah**
Baylor University, Waco, Texas
{paapa_quansah1, pablo_rivas, ernest_bonnah}@baylor.edu

## ABSTRACT

Bridging the gap between the formal precision of system specifications and the nuances of human language is critical for reliable engineering, robotics, and AI safety, but it remains a major bottleneck. Prior efforts in grounding formal logic remain fragmented, resulting in datasets that are very small-scale ($\sim 2 - 5k$ examples), domain-specific, or translate logic into overly technical forms rather than context-rich natural language (NL), failing to adequately bridge formal methods and practical NLP. To address this gap, we introduce **VERIFY**, the first large-scale dataset designed to unify these elements. This dataset contains more than **200k+** rigorously generated triplets, each comprising a Linear Temporal Logic (LTL) formula, a structured, human-readable **'Intermediate Technical Language' (ITL)** representation designed as a bridge between logic and text, and a domain-specific NL description contextualized across **13 diverse domains**. VERIFY's construction pipeline ensures high fidelity: LTL formulas are enumerated and verified via model checking, mapped to the novel ITL representation using a **provably complete** formal grammar, and then translated into context-aware NL via LLM-driven generation. We guarantee data quality through extensive validation protocols, including manual expert verification of 10,000 diverse samples. Furthermore, automated semantic consistency checks judged by Llama 3.3 confirmed an estimated **>97% semantic correctness**. From the initial experiments, we demonstrate VERIFY's scalability, logical complexity, and contextual diversity, significantly challenging standard models such as T5 and Llama 3.

 **Data & Code:** github.com/sedislab/Verify

 **Data & Dataset Card:** huggingface.co/datasets/sedislab/VERIFY

kaggle **Data & Dataset Card:** kaggle.com/datasets/sedislab/verify

## 1 INTRODUCTION

The increasing complexity of software systems, autonomous agents, and critical infrastructure, from financial trading algorithms and medical devices to aerospace controls and smart grids necessitates rigorous methods for specifying behavior and ensuring reliability (Boehm, 2002; Knight, 2002). Formal methods provide mathematically grounded techniques for specifying, developing, and verifying such systems (Clarke, 1997; Clarke et al., 2018; Holzmann, 2004). Temporal logics such as Linear Temporal Logic (LTL) (Pnueli, 1977) are now widely used for specifying and verifying properties of critical hardware, software, and communication systems (Rozier, 2011). For instance, consider a specification of a home automation system that requires that "*If any exterior door opens after 10 p.m., the security lights should immediately turn on and stay on until the door is closed.*" Using LTL formalism, this requirement can be expressed as $\mathbf{G}\big((t > 22:00 \land door\_open) \rightarrow lights\_on \mathbf{U} \neg door\_open\big)$. Despite this precision, the syntax and semantics of formal logic render specifications opaque to domain experts and many developers, limiting the adoption of formal methods in practice (Easterbrook et al., 1998; Dwyer et al., 1999). Conversely, system requirements are frequently documented in natural language (NL), which is accessible but prone to ambiguity, incompleteness, and inconsistency. This leads to specification misunderstandings and costly errors in safety-critical contexts (Berry et al., 2003; Hwang and Rine, 1998; Großer et al., 2022). Closing this gap requires large-scale resources that systematically align unambiguous formal expressions with their context-rich NL counterparts (Wing, 1990). Thus, there is the need to create large-scale resources that systematically align such unambiguous formal expressions with their context-rich NL counterparts.

Progress toward this goal has been hampered by the limitations of available datasets (Andreas et al., 2013; Jiang et al., 2024). Existing resources linking temporal logic and natural language typically fall short in several critical dimensions such as confinement to single, niche application domains—such as robotics commands (Scalise et al., 2018) or specific software verification patterns (Neyzov and Kuzmin, 2023), inhibiting the development of cross-domain, generalizable models.

To address this critical gap, we introduce **VERIFY**, a large-scale dataset that unifies three levels of representation; LTL formulas, a structured Intermediate Technical Language (ITL), and domain-specific natural language. VERIFY contains over 200 thousand rigorously generated triplets, each comprising: (i) a Linear Temporal Logic (LTL) formula specifying a temporal property, (ii) a structured, human-readable 'Intermediate Technical Language' (ITL) representation, novel to this work, explicitly designed to bridge the structural patterns of LTL with the syntax of natural language and (iii) a domain-specific Natural Language (NL) description expressing the property in context.

The construction of VERIFY prioritizes both scale and fidelity. LTL formulas are systematically enumerated and formally verified for non-triviality and satisfiability using model checking (Clarke, 1997). Each verified LTL formula is then mapped to our ITL representation using a formal grammar engineered to be provably complete with respect to the input LTL fragment. Finally, context-aware NL descriptions are generated using a state-of-the-art reasoning large language model (Liu et al., 2024; Guo et al., 2025), conditioned on the LTL/ITL structure, the domain, and domain-specific variable semantics. Crucially, data quality is guaranteed through extensive validation protocols: manual expert verification of 10,000 random samples and automated semantic consistency and correctness checks judged by an LLM-as-judge approach (using Llama 3.3) (Grattafiori et al., 2024; Son et al., 2024) across 18% of the dataset, confirming an estimated $> 97\%$ semantic integrity.

This work makes four primary contributions. First, we release *the VERIFY dataset*: over 200,000 LTL-ITL-NL triplets spanning 13 application domains. Second, we introduce *the ITL formalism*, a structured intermediate representation with a provably total mapping from LTL. Third, we describe a *multi-stage construction pipeline* incorporating model checking, formal grammars, LLM generation, and extensive human and automated validation. Fourth, we report *baseline experiments* with T5, BART, Llama 3, Mistral, CodeLlama, and DeepSeek Coder that establish performance benchmarks and reveal the difficulty of NL-to-LTL translation (Raffel et al., 2020; Grattafiori et al., 2024). The full dataset, ITL specification, and evaluation code are publicly released.

## 2 RELATED WORK

The challenge of bridging formal logical specifications and natural language descriptions is a long-standing pursuit with significant implications for system design, verification, requirements engineering, and human-robot interaction (Wing, 1990). This section reviews prior work relevant to the VERIFY dataset, focusing on approaches for translating between Linear Temporal Logic (LTL) and natural language (NL), the landscape of existing datasets, and the limitations that motivated VERIFY's development.

**Translating Between LTL and Natural Language:** Translating formal specifications like LTL into understandable natural language, and vice-versa, has been approached using various methods, ranging from rule-based systems to modern deep learning techniques. Early work on LTL-to-NL translation relied on attribute grammars and heuristics applied to the formula's parse tree (Brunello et al., 2019). However, achieving truly fluent, context-aware, and unambiguous translations that avoid common human misinterpretations (e.g., regarding temporal operators like "Until" and "Weak Until" (Greenman et al., 2022)) using purely rule-based methods remains an open challenge (Brunello et al., 2019). Translating NL to LTL using rule-based methods often involves complex semantic parsing pipelines, which can be incredibly brittle and difficult to scale (Kushman and Barzilay, 2013). With the rise of deep learning, Neural Machine Translation (NMT) approaches have been applied to LTL-NL translation. A seminal effort by Cherukuri et al. (Cherukuri et al., 2022) demonstrated the feasibility of using OpenNMT (Klein et al., 2017) to translate LTL formulas into English explanations, achieving high BLEU scores on a dataset augmented with variable permutations. This highlighted the potential of data-driven methods but also their dependence on sufficiently large and representative paired corpora. Similar sequence-to-sequence models have been explored for NL to LTL tasks, often framing it as a translation problem (Wang, 2020; Vaezipoor et al., 2021).

More recently, Large Language Models (LLMs) have shown significant promise. Pan et al. (Pan et al., 2023) employs GPT-3 for paraphrasing structured English templates (derived from LTL via

rules/templates) to synthesize diverse NL commands for training NL to LTL models data-efficiently. The Lang2LTL work also utilizes LLMs within its translation framework (Liu et al., 2023). The NL2TL project (Chen et al., 2023) uses GPT-3 to generate a large dataset of "lifted" NL-Temporal Logic pairs where specific details are abstracted away and fine-tunes T5 models, demonstrating LLMs' potential for both data generation and translation, particularly when aiming for cross-domain generalization via lifted representations. Tooling efforts like the NL2LTL Python package also integrate LLMs (GPT) alongside traditional NLU engines (Rasa) to translate NL into predefined LTL patterns. These works showcase the power of LLMs but often still rely on intermediate structures, specific patterns, or focus on lifted representations rather than fully contextual, grounded NL across truly diverse domains. Furthermore, work like Greenman et al.'s (Greenman et al., 2022) reminds us that effective translation requires more than semantic equivalence; it demands alignment with human cognitive patterns and expectations. Their user studies revealed systematic misunderstandings when humans map LTL to English, emphasizing that automated translation must produce outputs that align with both formal semantics and user expectations to be truly effective for explanation or requirements validation.

**Existing Datasets and Resources:** Despite progress in translation methodologies, a major bottleneck that remains is the availability of large, diverse, and suitable datasets. While several resources have been created, a review reveals significant limitations, especially concerning scale, domain diversity, contextual richness, and accessibility. A significant portion of publicly available datasets is confined to narrow application domains, primarily robotics and navigation. Examples include the datasets from Pan et al. (Pan et al., 2023), Wang et al. (Wang et al., 2021), the Language-to-Landmarks work (Berg et al., 2020), and the grounded parts of Lang2LTL (Liu et al., 2023). While valuable within their specific contexts, these resources lack the linguistic and conceptual diversity needed to train models that generalize beyond command-and-control scenarios. Other datasets operate at a symbolic level, using abstract variable names (Cherukuri et al., 2022; Liu et al., 2023), or focus on specialized areas like hardware verification (Hahn et al., 2022). While the NL2TL dataset (Chen et al., 2023) attempts cross-domain generalization using lifted representations, it differs from providing specific, contextual groundings across varied domains.

Furthermore, many existing datasets are limited in scale, often containing only a few thousand (Wang et al., 2021; Berg et al., 2020; Liu et al., 2023) or low tens of thousands (Cherukuri et al., 2022; Chen et al., 2023) of examples. This scale is often insufficient to train large neural models capable of capturing the complex interplay between logical structure and linguistic variation and serves for a light finetuning. Perhaps most critically, the nature of the natural language presented is often restricted. Many datasets feature imperative commands or relatively technical descriptions that closely mirror the underlying logic (Pan et al., 2023; Kushman and Barzilay, 2013), rather than the richer, more descriptive, and context-dependent language typically found in real-world requirements documents or system descriptions (Beltagy et al., 2020).

Finally, accessing and utilizing these resources can be challenging due to their fragmented nature, originating from different research groups with varying formats and objectives. So, despite valuable contributions within specific niches, a large-scale, multi-domain dataset featuring rich, contextual NL paired with LTL has been conspicuously absent.

**Where Does VERIFY Fit In:** VERIFY addresses these combined limitations. Its scale (200k+ triplets), domain breadth (13 areas), contextual NL grounding, and the novel ITL intermediate layer collectively distinguish it from all prior resources. Table 1 summarizes the comparison.

## 3 THE VERIFY DATASET

This section describes VERIFY's structure, its three-layer representation framework, and key dataset statistics.

### 3.1 CONCEPTUAL FRAMEWORK: UNIFYING LTL, ITL, AND CONTEXTUAL NL

VERIFY is built upon a three-layer representation; Linear Temporal Logic (LTL), an Intermediate Technical Language (ITL), and Natural Language (NL), all grounded within specific application domains and contexts.

**Linear Temporal Logic (LTL)**: At the core, LTL serves as the formal specification language (Pnueli, 1977). It allows precise expression of properties over time using propositional variables, standard

Table 1: Comparison of VERIFY with existing LTL-NL datasets. "Contextual" NL refers to descriptions grounded in domain-specific variable definitions, as opposed to generic templates or imperative commands. A dash indicates the property is absent or not applicable.

| Dataset | Scale | Domains | NL Type | Intermediate Repr. | Formal Verification | Public |
|---|---|---|---|---|---|---|
| Cherukuri et al. (Cherukuri et al., 2022) | ~12k | Abstract / symbolic | Technical | — | — | Partial |
| Pan et al. (Pan et al., 2023) | ~6.5k | Robotics | Commands | — | — | Yes |
| Wang et al. (Wang et al., 2021) | ~2k | Robotics | Commands | — | — | Yes |
| Berg et al. (Berg et al., 2020) | ~1.5k | Navigation | Commands | — | — | Yes |
| Lang2LTL (Liu et al., 2023) | ~2.1k | Robotics / navigation | Commands | — | — | Yes |
| NL2TL (Chen et al., 2023) | ~28k | Lifted (cross-domain) | Lifted templates | — | — | Yes |
| Hahn et al. (Hahn et al., 2022) | ~100k | Hardware verification | Technical | — | — | Yes |
| **VERIFY (ours)** | **200k+** | **13 diverse** | **Contextual** | **ITL** | **✓ (Spot)** | **Yes** |

boolean operators ($\neg, \wedge, \vee, \rightarrow$), and temporal modal operators. VERIFY utilizes the standard LTL operators: **G** ('Globally' or 'Always'), **F** ('Finally' or 'Eventually'), **X** ('Next'), **U** ('Until'), **W** ('Weak Until'), and **R** ('Release'). For instance, $\mathbf{G}(req \rightarrow \mathbf{F}(ack))$ formally states that it is always the case that if a request $req$ is sent, then an acknowledgment $ack$ must be sent back at some point in the future. This layer provides the unambiguous, machine-verifiable meaning.

**The Intermediate Technical Language (ITL)**: A key challenge in this whole research area is the significant semantic gap between the abstract, symbolic nature of LTL and the rich, nuanced, context-dependent nature of real-world natural language. Directly mapping complex LTL formulas to fluent, accurate, and contextual NL is extremely difficult. This is because translations often become overly technical, template-like, or lose semantic fidelity. To address this, we introduce ITL, a novel intermediate representation designed specifically to serve as a structural and semantic bridge between LTL and NL. ITL is conceived to be more structured and less ambiguous than free-form NL, yet more human-readable and linguistically closer to NL than raw LTL formulas. ITL preserves the logical structure of the LTL formula's abstract syntax tree while replacing symbolic operators with controlled English keywords drawn from a curated library of human-readable temporal expressions. This intermediate target simplifies the translation task in both directions (LTL $\rightarrow$ NL and NL $\rightarrow$ LTL). Given the LTL formula: $\mathbf{G}(system\_ready \rightarrow (check\_a \ \mathbf{U} \ check\_b))$. An ITL representation might be: **Always**(IF $system\_ready$ THEN ($check\_a$ Until $check\_b$)) This ITL form preserves the exact logical structure (**always, implies, until**) but uses more verbose, keyword-like operators, making the transition to or from NL more manageable than directly handling the symbolic LTL.

**Domain and Context**: While LTL provides formal meaning and ITL offers a structural bridge, generating truly relevant NL requires grounding in a specific application context. An LTL formula like $\mathbf{G}(p \rightarrow \mathbf{F}q)$ is abstract; its meaningful NL translation depends entirely on what $p$ and $q$ represent in a given scenario. VERIFY incorporates this crucial grounding through two key fields associated with each triplet: (i) *domain*: Specifies the application area (e.g., 'Financial Services', 'Home Automation') and (ii) *activity*: Provides natural language definitions for the propositional variables used in the LTL formula within that domain's context (e.g., $p$ = user login attempt succeeds, $q$ = two-factor authentication prompt is displayed).

This domain and activity information provides the essential semantic context, enabling the generation and interpretation of NL descriptions that are not generic templates but are instead specific, relevant, and interpretable within their intended domain. For instance, grounded in a financial domain, the ITL above might translate to: "It must always be the case that if the trading system reports ready, then check A must remain valid until check B is completed."

## 3.2 DATASET DESIGN AND STRUCTURE

The creation of VERIFY was guided by several core principles aimed at producing a high-quality, impactful resource for the research community. We aimed for Logical Diversity, ensuring the dataset includes a wide spectrum of LTL formulas, varying in structure, operator usage, and nesting depth. Contextual Richness was a primary design goal; driving the generation of NL that is deeply specific to the domain and variable definitions, avoiding vague or purely syntactic translations. Broad Domain Coverage across 13 distinct areas was incorporated to facilitate research into domain generalization and adaptation. Verifiability and Quality were central, addressed through formal verification of

LTL formulas, a provably correct LTL-to-ITL mapping, and multi-stage validation of NL alignment (detailed in Section 4). Finally, Scalability was a key goal, resulting in a large-scale dataset suitable for training modern deep learning models.

**Data Schema**: The dataset is structured as a collection of records, where each record represents a complete LTL-ITL-NL triplet with its associated context and metadata. The primary fields are described in Table 15.

**Domain Coverage**: VERIFY spans 13 distinct application domains, selected to cover a wide range of scenarios where formal specification and natural language descriptions interact. The domains are listed in Table 13.

### 3.3 DATASET STATISTICS

VERIFY is a large-scale resource comprising over 200 thousand LTL-ITL-NL triplets. This includes a substantial number of unique LTL formulas and ITL structures, reflecting diverse logical patterns. Due to the contextual generation process, the vast majority of the 200k+ NL translations are unique or near-unique within their specific domain context. The dataset features a broad distribution of LTL formula complexities, ranging from simple properties involving one or two operators to complex specifications with significant nesting depths. The Natural Language descriptions also exhibit variety. Sentence lengths vary considerably depending on the complexity of the underlying logic and the specific domain context. The sample distribution based on the count of temporal operators per formula and complexity of the underlying logic and the specific domain context is shown in Appendix A. We also show the sample distribution across the specific domains in the Appendix A.

## 4 DATASET CONSTRUCTION METHODOLOGY

The creation of the VERIFY dataset involved a rigorous, multi-stage pipeline designed to create high-quality data encompassing LTL, ITL, and contextual NL. This process emphasized formal correctness, semantic consistency, contextual relevance, and scalability, incorporating automated generation, formal verification, LLM capabilities, and comprehensive quality assurance steps.

### 4.1 LTL FORMULA GENERATION AND VERIFICATION

The foundation of VERIFY lies in a diverse set of syntactically correct and semantically meaningful LTL formulas.

We employed a programmatic LTL formula enumerator that recursively constructs formulas up to a specified maximum depth (depth 25 in our process) using standard LTL operators (**G, F, X, U, R, W**) and boolean connectives, applied to a set of atomic propositions ($p$ through $w$). The generation depth of 25 refers to the maximum recursion depth of the formula enumerator, which controls the size of the search space. The AST depth of formulas retained after verification and canonicalization is substantially lower (mean 5.0; see the per-domain statistics in Table 14), because Spot's canonicalization simplifies many deeply nested structures into equivalent shorter forms. Random choices at each step ensure structural diversity in the generated formulas. To manage the vast number of potential formulas and avoid trivial duplicates, generated formulas undergo a structural canonicalization process. This involves conversion to Negation Normal Form (NNF), expansion of implications/equivalences, application of associative/distributive laws, and sorting of operands for commutative operators. A canonical hash is computed for each unique structure. Formulas are stored in an SQLite database indexed by this hash, so that only structurally distinct formulas are retained.

The primary step ensuring the semantic validity and non-triviality of the LTL formulas involves rigorous verification using Spot (Duret-Lutz and Poitrenaud, 2004). This critical step filters out syntactically invalid formulas that might arise from the generator or initial conversion. Successfully validated formulas, along with their canonical string representation as determined by Spot are stored back into the database (spot_formulas and canonical_form columns). This ensures that all LTL formulas used in subsequent stages are well-formed and provides a standardized representation grounded in a formal verification tool.

### 4.2 INTERMEDIATE TECHNICAL LANGUAGE (ITL) GENERATION AND VERIFICATION

The generation of ITL from verified LTL formulas is a deterministic, rule-based process. Verified LTL formulas are first parsed into an Abstract Syntax Tree (AST) representation using Spot's parsing

capabilities. This captures the precise logical and temporal structure. An AST-visitor script then walks the LTL parse tree and, at each operator node, performs an $O(1)$ dictionary lookup of a curated list of human-readable templates. These templates are derived from a curated grammar based on common human-readable expressions for temporal concepts. For example, $\mathbf{G}(p)$ maps to Always $p$, and $p\mathbf{U}q$ maps to $p$ Until $q$. This recursive process yields a canonical ITL string that directly mirrors the LTL structure but uses more naturalistic keywords.

**ITL Verification**: To ensure the integrity of the ITL generation process and the semantic equivalence between the canonical ITL and its source LTL, an automated verification step was implemented. This involves parsing the ITL text back into an LTL formula representation using a rule-based parser guided by the ITL grammar via an AST. This reconstructed LTL formula is then formally compared against the original LTL using Spot's built-in semantic equivalence checker. This check confirms that the ITL representation, when interpreted back through its grammar rules, retains the precise logical meaning of the source LTL. Because the ITL generation strictly follows the LTL AST structure and uses a defined mapping for each LTL operator, the transformation from LTL to the canonical ITL output is deterministic and structure-preserving. This deterministic mapping forms the basis of the provably complete relationship between the source LTL and the canonical ITL representation. We prove this relationship in Appendix C.

### 4.3 Contextual Natural Language (NL) Generation

To generate natural language descriptions relevant to specific application areas, we utilized a large language model guided by the LTL formula, its ITL representation, and domain context. We used DeepSeek-R1 (Liu et al., 2024; Guo et al., 2025) for NL generation. For each LTL/ITL pair selected from the database, a target domain was chosen using a probabilistic sampling strategy designed to balance the distribution across the 13 domains. A prompt (detailed in Appendix E.3) was constructed, instructing the LLM to act as an expert in formal methods and the target domain. The prompt provided the LTL formula and the ITL representation and explicitly requested the model to generate two components within specific tags: ***<activity>***: A natural language description defining the meaning of the atomic propositions within the context of the selected domain. ***<translation>***: A clear, concise, and semantically accurate natural language translation of the LTL/ITL logic, incorporating the domain context provided in the ***<activity>*** tag.

### 4.4 Quality Assurance and Validation

Quality assurance involved multiple stages targeting different components of each triplet. As described above, LTL formula validity is enforced through parsing and canonicalization using the Spot library. The canonical ITL is generated via a deterministic, structure-preserving mapping from the verified LTL, ensuring structural fidelity by construction. Even then, the consistency between ITL and the original LTL was further verified using an ITL-to-LTL parser and Spot-based equivalence checking.

**Manual NL Check**: A significant manual review was conducted on the completed dataset. We randomly sampled 10,000 LTL-ITL-NL triplets and reviewed them. The review focused on: (i) Semantic Equivalence: Does the NL translation accurately convey the precise meaning of the LTL/ITL formula, especially the temporal relationships? (ii) Contextual Relevance: Is the activity description plausible for the domain, and is the translation consistent with this context? and (iii) Linguistic Quality: Is the NL translation fluent, grammatically correct, and easily understandable? This manual check identified a very low error rate (<1%), primarily consisting of minor fluency issues or occasional subtle deviations in temporal meaning, which were used to refine prompts and generation strategies iteratively.

**LLM Judge (NL)**: To augment manual checks and provide broader validation coverage, we employed an automated LLM-based judge. We utilized Llama 3.3 70B Instruct (Grattafiori et al., 2024) to evaluate a random sample making up a total of 18% of the generated NL translations. The LLM judge was presented with the LTL formula, ITL text, and the NL translation. It was prompted to assess semantic precision (especially regarding temporal operators), contextual appropriateness, and fluency, outputting a structured JSON response containing: is_correct (boolean), score (0-10 integer rating), issues (a list of identified problems), and textual reasoning for its judgment. The results from the LLM judge indicated an estimated >97% semantic correctness and consistency between the NL translations and their corresponding LTL/ITL specifications, aligning closely with the findings from

Table 2: Performance on LTL/ITL-to-NL translation tasks (BERTScore F1 / ROUGE-L F1).

| Model | LTL $\rightarrow$ NL | ITL $\rightarrow$ NL |
|---|---|---|
| T5-base | 0.62 / 0.37 | 0.84 / 0.41 |
| T5-large | 0.67 / 0.41 | 0.89 / 0.61 |
| BART-large | 0.63 / 0.39 | 0.78 / 0.56 |
| Llama-3-8B-Instruct (FT) | 0.91 / 0.67 | 0.94 / 0.73 |
| Mistral-7B-Instruct (FT) | 0.88 / 0.62 | 0.91 / 0.62 |
| CodeLlama-7B-Instruct (FT) | 0.88 / 0.63 | 0.92 / 0.71 |

the manual verification phase. We note that because both the generator (DeepSeek-R1) and the judge (Llama 3.3) are LLMs, errors that one model makes may be systematically missed by the other.

## 5 BENCHMARK TASKS AND EXPERIMENTS

To demonstrate the utility of VERIFY and establish baseline performance levels for future research, we conducted a set of experiments to evaluate state-of-the-art models on core translation tasks enabled by VERIFY's LTL-ITL-NL structure and probe the challenges introduced by its contextual richness, domain diversity, and logical complexity.

### 5.1 EXPERIMENTAL SETUP

We created standardized train, validation, and test splits for the VERIFY dataset, maintaining an approximate 80%/10%/10% ratio. These splits were stratified by domain to ensure representation across all 13 areas in each set.

**Baseline Models**: We evaluated a diverse range of models to provide a broad performance landscape; (i) **Pre-trained Sequence-to-Sequence Models**: Standard Transformer-based models, specifically T5 (`t5-base`, `t5-large`) (Raffel et al., 2020) and BART (`bart-base`, `bart-large`) (Lewis et al., 2019), were fine-tuned for each task, (ii) **Instruction-Tuned LLMs**: We fine-tuned prominent instruction-following models, including Llama 3 (`Llama-3-8B-Instruct`) (Grattafiori et al., 2024) and Mistral (`Mistral-7B-Instruct-v0.2`) (Jiang et al., 2023), to assess the capabilities of modern LLMs on these structured tasks, and (iii) **Code-Focused LLMs**: Models pre-trained extensively on code, such as CodeLlama (`CodeLlama-7b-Instruct-hf`) (Rozière et al., 2024) and DeepSeek Coder (`deepseek-coder-6.7b-instruct`) (Guo et al., 2024), were included, particularly for tasks involving generation of formal LTL/ITL outputs. All models were fine-tuned using standard hyperparameters optimized on the validation set (details in Appendix A).

**Evaluation Metrics**: We employed a suite of metrics appropriate for the different translation directions: (i) **NL Generation (LTL/ITL $\rightarrow$ NL)**: Primary metrics were BERTScore (Zhang et al., 2019) (for semantic similarity using DeBERTa-v3-large) and ROUGE-L (Lin, 2004) (for lexical overlap) and (ii) **Logic Generation (NL $\rightarrow$ LTL/ITL, LTL $\leftrightarrow$ ITL)**: Primary metrics were task-dependent: Semantic Equivalence (for NL $\rightarrow$ LTL and ITL $\rightarrow$ LTL, using Spot to check logical equivalence with the ground truth) and Exact Match (EM) (especially for NL $\rightarrow$ ITL and LTL $\rightarrow$ ITL). Secondary metrics included Tree Edit Distance (TED) to measure structural similarity, and Syntactic Correctness (percentage of outputs parsable according to the LTL/ITL grammar).

### 5.2 CORE TRANSLATION TASK PERFORMANCE

Tasks 1 & 2 (LTL/ITL $\rightarrow$ NL): Generating contextual natural language from formal (LTL) or intermediate (ITL) representations, given domain and activity context. Results (Table 2) show that modern pre-trained models achieve reasonable performance, with LLMs generally outperforming T5/BART, particularly on semantic metrics like BERTScore. Generating NL from ITL yields comparable or slightly better results than from LTL directly for most models, suggesting ITL can be an effective input representation. Tasks 3 & 4 (NL $\rightarrow$ LTL/ITL): As expected, these tasks proved significantly more challenging (Table 3). Semantic Equivalence for NL$\rightarrow$LTL remains low across models, highlighting the difficulty of precise logical form recovery from ambiguous NL. We also do recognize that semantic equivalence as measured by Spot is a strict binary criterion: two formulas that differ by a single operator score zero even if they are otherwise structurally identical. The low absolute

numbers in Table 2 therefore reflect both genuine model failures and the severity of the metric. Future work might supplement semantic equivalence with partial-credit measures such as tree edit distance on the formula ASTs to better characterize the distribution of errors. Code-focused LLMs showed a slight advantage in generating syntactically correct outputs. Exact Match for NL→ITL was higher than for LTL, potentially due to ITL's more constrained structure, but still far from perfect.

Table 3: Performance on NL-to-LTL/ITL translation tasks (Semantic Equiv. / EM / Syntactic Correctness.)

| Model | NL → LTL (SemEq / EM / SynCorr) | NL → ITL (EM / TED / SynCorr) |
|---|---|---|
| T5-large | 22.3 / 2.8 / 66.1 | 2.2 / 11.8 / 68.3 |
| Llama-3-8B-Instruct (FT) | 28.2 / 4.1 / 73.6 | 4.3 / 23.5 / 77.2 |
| Mistral-7B-Instruct (FT) | 25.6 / 2.9 / 68.4 | 1.6 / 17.9 / 74.5 |
| CodeLlama-7b-Instruct (FT) | 25.4 / 3.3 / 71.1 | 3.2 / 19.2 / 74.8 |
| DeepSeek-Coder (FT) | 31.5 / 5.4 / 74.2 | 4.1 / 18.8 / 79.5 |

Task 5 (LTL ↔ ITL): Translating directly between the formal LTL (Spot canonical form) and the canonical ITL. Given the deterministic rule-based mapping used for canonical ITL generation, models achieved Exact Match scores up to 31.7% on LTL→ITL (Table 4). The ITL→LTL direction also showed high Semantic Equivalence (up to 56.4%) and corresponding Exact Match scores (up to 21.6%), confirming models can effectively learn the structural correspondence, although minor syntactic variations occasionally occurred.

Table 4: Performance on LTL↔ITL translation tasks (Exact Match / Semantic Equiv.)

| Model | LTL → ITL (EM) | ITL → LTL (SemEq / EM) |
|---|---|---|
| T5-large | 19.3 | 38.6 / 19.0 |
| Llama-3-8B-Instruct (FT) | 31.7 | 53.1 / 20.8 |
| CodeLlama-7b-Instruct (FT) | 27.9 | 56.4 / 21.6 |

**Error Analysis.** To understand *why* models fail at NL→LTL translation, we manually reviewed 100 incorrect predictions from the Llama-3-8B-Instruct (FT) model on Task 3. Each prediction was classified into one of five error categories based on the primary failure mode. Table 5 summarizes the results.

Table 5: Error analysis of 100 incorrect NL→LTL predictions from Llama-3-8B-Instruct (FT). Categories are mutually exclusive; each prediction is assigned to its primary failure mode.

| Error Category | Freq. | Description |
|---|---|---|
| Incorrect Logical Scope | 41% | Misplaced parentheses or incorrect nesting of operator scope. |
| Temporal Operator Mismatch | 28% | Confusion between semantically close operators, most often U/W or G/F. |
| Propositional Atom Error | 17% | Missing or hallucinated propositional atoms relative to the activity context. |
| Contextual Grounding Failure | 9% | Logically valid LTL that does not match the variable definitions in the activity field. |
| Syntactic Malformation | 5% | Output cannot be parsed as a well-formed LTL formula. |

Syntactic errors account for only 5% of failures, indicating that fine-tuned LLMs reliably learn the surface grammar of LTL. The dominant failure mode is incorrect logical scope (41%), where the model produces a parseable formula whose operator nesting does not match the intended reading of the source sentence; for instance, generating $(\mathbf{G}(p \rightarrow q))\ \mathbf{U}\ r$ instead of $\mathbf{G}(p \rightarrow (q\ \mathbf{U}\ r))$.

Temporal operator mismatch (28%) is the second most common failure: the model frequently substitutes Until (U) for Weak Until (W), conflating a liveness requirement with a safety property, a distinction that English "until" leaves ambiguous but that carries significant formal consequences. Propositional atom errors (17%) and contextual grounding failures (9%) together suggest the model loses track of the mapping between domain-specific variable names and abstract atoms, consistent with the sharp performance drop observed when context is removed (Table 6). A fuller discussion appears in Appendix B.3.

## 5.3 ANALYTICAL EXPERIMENTS

We performed further experiments to analyze the influence of VERIFY's specific design choices and characteristics.

**Experiment A (Value of ITL)**: We investigated the potential of ITL as an effective intermediate representation for logic-to-NL generation. This was assessed by comparing the performance of direct LTL → NL translation (Task 1) with the performance of ITL → NL translation (Task 2), which represents the second stage of a potential two-stage pipeline (LTL → ITL via Task 5 model, then ITL → NL via Task 2 model). The results, illustrated in Figure 1, demonstrate that models consistently achieve higher performance when translating ITL to NL compared to translating LTL to NL. Specifically, for all evaluated models, both BERTScore F1 and ROUGE-L F1 scores are notably improved when ITL is the source language for NL generation as opposed to LTL. For instance, Llama-3-8B-Instruct (FT) achieves a BERTScore F1 of 0.91 for LTL → NL, which increases to 0.94 for ITL → NL; similarly, its ROUGE-L F1 improves from 0.67 to 0.73. T5-large shows an even more pronounced relative improvement in BERTScore F1, jumping from 0.67 (LTL → NL) to 0.89 (ITL → NL). These findings support the hypothesis that ITL can serve as a beneficial intermediate representation, as translating from ITL to NL yields significantly better semantic accuracy and lexical overlap than direct translation from LTL to NL across a range of models.

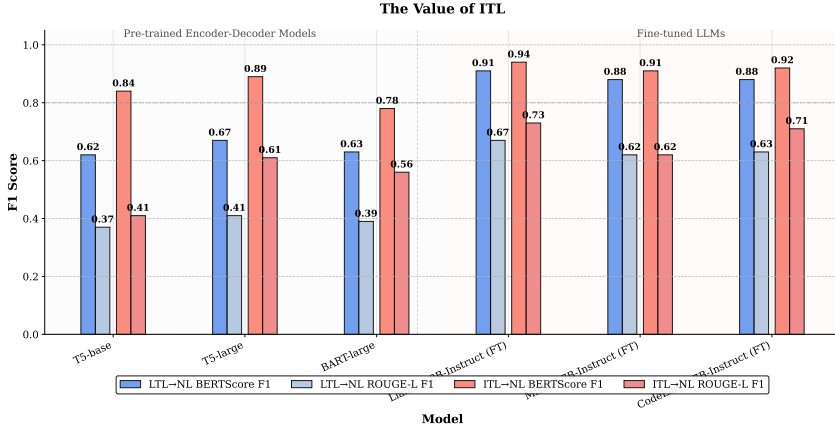

Figure 1: Comparison of direct LTL-to-NL translation versus a two-stage LTL-to-ITL-to-NL pipeline, showing difference in BERTScore vs. LTL complexity.

**Experiment B (Impact of Context)**: Models trained for NL → LTL/ITL translation without the domain and activity context information suffered a significant performance degradation compared to models trained with full context (Table 6). Semantic scores dropped considerably and Syntactic Correctness was much lower, confirming the importance of contextual grounding for generating meaningful and accurate translations. Note that Semantic Equivalence for NL→ITL in Table 6 is computed by round-tripping the predicted ITL through the ITL-to-LTL parser and checking logical equivalence against the ground-truth LTL via Spot; this metric is substantially higher than Exact Match because many syntactically distinct ITL strings map to semantically equivalent LTL formulas. Our experiments establish initial baselines on the VERIFY dataset, demonstrating the capabilities and limitations of current models on contextual logic-to-language tasks. While modern pre-trained models achieve strong performance on LTL/ITL → NL generation and inter-formalism translation (LTL ↔ ITL), significant challenges remain, particularly in parsing NL to accurate LTL specifications (NL → LTL) and generalizing across diverse domains. The results confirm the importance of contextual

Table 6: Impact of domain and activity context on NL-to-LTL/ITL translation performance (Semantic Equiv. / EM).

| Llama 3 FT | With Context | Without Context |
|---|---|---|
| NL → LTL | 28.2 / 4.1 | 7.7 / 0.8 |
| NL → ITL | 41.5 / 4.3 | 13.9 / 1.3 |

information, the potential utility of ITL for complex formulas, and VERIFY's effectiveness in representing a wide spectrum of logical and domain-based difficulties suitable for driving future research.

**Additional Experiments.** We conducted additional experiments to support the need for such a dataset; namely, per-domain performance analysis, generalization which was tested by holding out individual domains during training and by evaluating the model on an entirely unseen formal logic (STL). Finally, a manual error analysis was performed on incorrect NL to LTL translations to categorize common failure patterns B.

## 6    CONCLUSION

Our experiments establish baselines on VERIFY that reveal an asymmetry in current model capabilities. Generating fluent NL from LTL or ITL is achievable with fine-tuned LLMs, which reach BERTScore F1 above 0.91. The reverse direction remains far harder: the best model achieves only 31.5% semantic equivalence on NL→LTL, with error analysis (Appendix B.3) attributing 41% of failures to incorrect operator scope and 28% to temporal operator confusion. These error patterns suggest that unconstrained autoregressive decoding is poorly suited to producing well-scoped logical formulas from natural language, and that structured decoding methods or neurosymbolic architectures may be needed.

The ITL layer provides a measurable benefit. Models translating ITL to NL consistently outperform those translating LTL to NL, with T5-large showing the largest relative gain (BERTScore F1 from 0.67 to 0.89). This supports the use of intermediate representations as a general strategy for bridging formal and informal languages, an idea with implications beyond LTL. VERIFY is publicly available under CC BY 4.0 at the repositories listed on the first page.

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

# A  DATASET STATEMENT

This statement provides details regarding the curation, content, potential risks, and administration of the VERIFY dataset, following recommended guidelines for dataset documentation.

## A.1  CURATION RATIONALE

VERIFY was created to address the limitations of existing LTL-NL datasets described in Sections 1 and 2. This appendix provides additional documentation following recommended dataset reporting guidelines. Figure 2(a) illustrates the distribution of VERIFY based on the count of temporal operators per formula. The Natural Language descriptions also exhibit variety. Sentence lengths vary considerably depending on the complexity of the underlying logic and the specific domain context, as shown in Figure 2(b). The overall NL vocabulary is extensive, reflecting the diverse terminology across the 13 domains. The sample distribution across the specific domains is shown in 2(c).

The goal is to provide a foundational resource to accelerate research in robust logic-to-language translation, formally-grounded NLP, domain adaptation for specifications, and human-centric verification, moving beyond niche applications or purely symbolic translations towards more realistic scenarios.

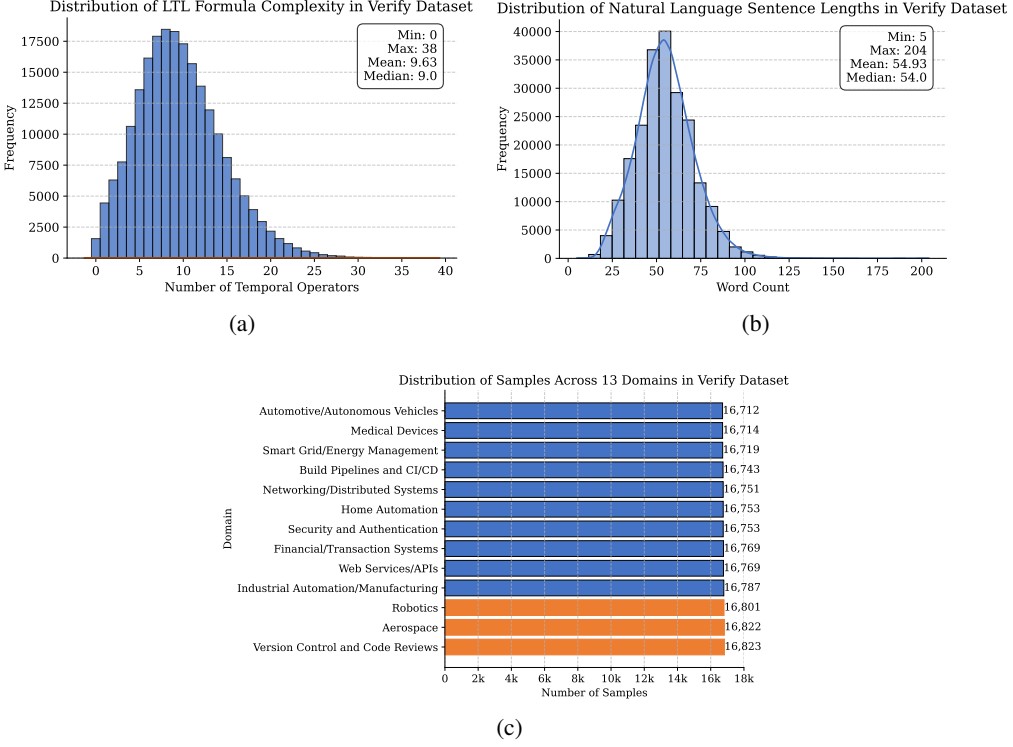

Figure 2: (a) Distribution of LTL formula complexity (temporal operator count) in VERIFY, showing coverage from simple to complex formulas (b) Distribution of Natural Language sentence lengths (by word count) in VERIFY (c) Distribution of samples across the 13 domains in VERIFY

## A.2  LANGUAGE VARIETY

The dataset contains three primary language types:

**LTL**: Standard Linear Temporal Logic formulas using common operators (G, F, X, U, R, W) and boolean connectives over atomic propositions (p-w). Formulas are stored in a canonical representation derived from the Spot library. **ITL**: A structured, rule-based Intermediate Technical Language designed for this dataset, using English keywords and templates corresponding to LTL operators. **NL**: Natural Language (English). The NL component was generated using a large language model

(DeepSeek-R1), prompted to produce context-specific descriptions based on the LTL/ITL structure and domain information. The vocabulary reflects the diversity of the 13 target domains (Table 13).

### A.3 SPEAKER/ANNOTATOR DEMOGRAPHICS

**LTL/ITL Generation**: These components were generated programmatically based on formal rules and algorithms. There were no human speakers or annotators directly involved in their generation, beyond the initial design of the ITL grammar rules and templates. **NL Generation**: The natural language descriptions were generated entirely by the DeepSeek-R1 large language model. No human speakers were involved. **Manual Validation (10k Sample)**: The manual check of 10,000 NL translations was performed by the paper's authors, all possessing graduate-level expertise in formal methods, temporal logic, and/or natural language processing. [All three are based at the same university]. No other demographic information was collected for this internal check. **LLM Judge Validation (18% Sample)**: The automated validation used the Llama 3.3 70B Instruct model. No human annotators were involved in this specific validation step.

### A.4 POTENTIAL RISKS & BIASES

**LLM Artifacts**: The primary risk stems from the use of LLMs for NL generation (DeepSeek-R1) and validation (Llama 3.3). While significant validation was performed, the generated NL may contain subtle stylistic biases, repetitive patterns, or occasional factual inconsistencies (particularly if the LLM struggled generating plausible activity descriptions) inherent in the foundation models used. The dataset might not fully capture the diversity and sometimes "ungrammatical" or ambiguous nature of truly human-generated requirements text. Note that the prompts used are in E.3. **Semantic Fidelity**: Although validation estimated >97% semantic correctness, subtle errors in translating complex temporal nuances might exist in a small fraction of the NL examples. Users should be aware that models trained on this data might inherit these subtle inaccuracies. **Scope Limitations**: The LTL formulas are generated up to a certain complexity (depth 25) and within a specific fragment; the dataset might not cover extremely complex or esoteric LTL patterns found in some specialized verification domains. The 13 domains, while diverse, are not exhaustive. **Misuse Potential**: Models trained on VERIFY could potentially be misused to generate plausible-sounding but incorrect natural language descriptions of formal properties, or conversely, misleading "formal-looking" specifications from ambiguous text, potentially obfuscating errors in critical systems if deployed without due diligence. **Mitigation**: We employed multi-stage validation (Spot verification for LTL, rule-based generation and equivalence checks for ITL, extensive manual and LLM-based checks for NL) to minimize errors. The dataset, code, and methodology are released openly to allow scrutiny. We encourage responsible use and awareness of these limitations.

### A.5 LIMITATIONS

We acknowledge several limitations inherent in VERIFY's current form. The natural language translations, while extensively validated, were primarily generated by an LLM (DeepSeek-R1); consequently, they may reflect the stylistic biases or occasional artifacts characteristic of such models and might not encompass the full spectrum of human linguistic variation for expressing logical concepts. The scope of LTL formulas, while diverse, was generated programmatically up to a certain complexity threshold (depth 25), and may not cover all possible patterns found in highly specialized specifications. Similarly, while the 13 domains offer broad coverage, they are not exhaustive of all potential application areas. Finally, the dataset primarily contains at most three canonical ITL and two contextual NL instance per LTL formula per domain context, limiting exploration of paraphrase diversity for now.

Despite these limitations, VERIFY opens numerous avenues for future research. Its scale and structure invite the development of novel model architectures specifically designed for formal logic translation, perhaps explicitly modeling the relationships between the three representations. The multi-domain nature makes it an ideal testbed for advancing few-shot and zero-shot domain adaptation techniques applied to formal specifications. Furthermore, the ITL layer could be investigated as a component in building more explainable AI systems for formal verification, potentially offering human-readable justifications derived from formal proofs. Extensions to multi-lingual contexts, generating NL in

languages other than English, represent another promising direction. The dataset could also inform the creation of more robust interactive tools for requirements elicitation and formalization.

The broader impact of this work lies in its potential to make formal methods more accessible and reliable. By facilitating better tools for translating between formal specifications and the natural language used by engineers, designers, and stakeholders, VERIFY can contribute to improved requirements engineering, reduced ambiguity, and ultimately, safer and more dependable systems in critical areas like aerospace, medicine, and finance. However, ethical considerations remain. The reliance on LLMs (DeepSeek-R1 for generation, Llama 3.3 for validation) means potential biases inherent in these models could be reflected in the dataset, despite mitigation through validation. Researchers using VERIFY should be mindful of these potential biases. Furthermore, while intended to improve clarity, models trained on this data could potentially be misused to generate misleading "formal-looking" requirements if not deployed responsibly. We encourage users to leverage the dataset's openness and rigorous validation framework for responsible innovation.

VERIFY addresses the long-standing challenge of grounding formal temporal logic in diverse, contextual natural language at scale. By providing over 200 thousand verified LTL-ITL-NL triplets across 13 domains, generated through a rigorous methodology incorporating formal checks and extensive validation, VERIFY offers a unique and valuable resource. VERIFY provides the foundational resource to spur significant advancements in formally-grounded natural language processing, enhance the synergy between the formal methods and NLP communities, and ultimately contribute to building more reliable and human-understandable complex systems. We release VERIFY openly and encourage the research community to utilize and extend this dataset to push the boundaries of logic-aware language understanding and generation.

### A.6 LICENSE

The VERIFY dataset is released under the Creative Commons Attribution 4.0 International (CC BY 4.0) license. The accompanying code is released under the MIT license. Please consult the respective license files in the repository for full details.

### A.7 MAINTENANCE PLAN

The VERIFY dataset will be hosted on Hugging Face Datasets, Kaggle Datasets and GitHub. We plan to maintain the dataset by addressing issues (e.g., errors, inconsistencies) reported by the community via the GitHub repository's issue tracker or direct contact with the authors. Updates or corrections will be managed through versioning on the Hugging Face Hub. While long-term active development beyond initial corrections is not guaranteed, we aim to keep the resource accessible and address critical issues for at least two years post-publication.

### A.8 DATASET USAGE EXAMPLES

The dataset is provided in standard CSV and Parquet formats for ease of use. Each record contains the LTL formula, canonical ITL, domain, activity context, and NL translation, along with identifiers and metadata (see Table 15). Users can load the data using standard libraries like Pandas or Hugging Face datasets. Example usage scripts and baseline model implementations are provided in the attached supplementary material but upon acceptance will be released to the general public to facilitate research on the tasks described in Section 5.

## B ADDITIONAL EXPERIMENTS

### B.1 PER-DOMAIN ANALYSIS OF THE DOMAINS

A detailed per-domain analysis is necessary for a multi-domain dataset and we have conducted a set of extensive per-domain analysis. We evaluated the performance of our Llama-3-8B-Instruct (FT) model on key translation tasks for all 13 domains. The results, presented in the table below, showcase the performance variations and highlight domain-specific challenges. Task 1 (LTL $\rightarrow$ NL) and Task 2 (ITL $\rightarrow$ NL) evaluated with BERTScore F1. Task 3 (NL $\rightarrow$ LTL) evaluated with Semantic Equivalence (SemEq %).

Table 7: Model Performance Across Various Domains

| Domain | Task 1 (BERTScore F1) | Task 1 (ROUGE-L) | Task 2 (BERTScore F1) | Task 2 (ROUGE-L) |
|---|---|---|---|---|
| Aerospace | 0.91 | 0.68 | 0.94 | 0.74 |
| Automotive/Autonomous Vehicles | 0.92 | 0.69 | 0.95 | 0.75 |
| Build Pipelines and CI/CD | 0.90 | 0.66 | 0.93 | 0.72 |
| Financial/Transaction Systems | 0.90 | 0.65 | 0.93 | 0.71 |
| Home Automation | 0.93 | 0.71 | 0.95 | 0.76 |
| Industrial Automation/Manufacturing | 0.92 | 0.68 | 0.94 | 0.73 |
| Medical Devices | 0.91 | 0.67 | 0.94 | 0.73 |
| Networking/Distributed Systems | 0.90 | 0.66 | 0.93 | 0.71 |
| Robotics | 0.92 | 0.70 | 0.95 | 0.75 |
| Security and Authentication | 0.91 | 0.68 | 0.94 | 0.72 |
| Smart Grid/Energy Management | 0.91 | 0.67 | 0.93 | 0.72 |
| Version Control and Code Reviews | 0.90 | 0.67 | 0.93 | 0.73 |
| Web Services/APIs | 0.91 | 0.68 | 0.94 | 0.74 |

Table 8: Model Performance for Task 3

| Domain | Task 3 (SemEq (%)) |
|---|---|
| Aerospace | 27.5 |
| Automotive/Autonomous Vehicles | 28.9 |
| Build Pipelines and CI/CD | 26.1 |
| Financial/Transaction Systems | 25.8 |
| Home Automation | 30.1 |
| Industrial Automation/Manufacturing | 28.2 |
| Medical Devices | 27.9 |
| Networking/Distributed Systems | 26.5 |
| Robotics | 29.5 |
| Security and Authentication | 27.1 |
| Smart Grid/Energy Management | 27.3 |
| Version Control and Code Reviews | 26.8 |
| Web Services/APIs | 27.0 |

This analysis already reveals important trends. For generation tasks (Tasks 1 and 2), all domains achieve high BERTScore and ROUGE-L scores, with Home Automation showing slightly better performance. For the more challenging translation from NL to a formal representation (Task 3), performance varies more significantly. Home Automation again leads, achieving a Semantic Equivalence of 30.1% in the NL→LTL task. In contrast, the Financial/Transaction Systems domain, which often involves more abstract concepts and complex causal relationships, proves more difficult for the model, resulting in the lowest scores for semantic equivalence. This suggests that the abstract nature of a domain's language directly impacts the difficulty of grounding it in formal logic.

Furthermore, we analyzed how performance is affected by the logical complexity of the LTL formulas, using AST depth as a proxy. The table below shows that as the formula depth increases, model performance on the most challenging NL→LTL task degrades noticeably.

Table 9: Semantic Equivalence by LTL Formula AST Depth

| LTL Formula AST Depth | Semantic Equivalence (%) |
|---|---|
| 1-4 (Low Complexity) | 35.4 |
| 5-8 (Medium Complexity) | 28.1 |
| 9-12 (High Complexity) | 21.9 |
| 13+ (Very High Complexity) | 15.2 |

The specific LTL depths used to define the categories are as follows: Low Complexity (depths 1–4), Medium Complexity (5–8), High Complexity (9–12), and Very High Complexity (13+). For each of the four complexity categories, we randomly sampled 1,000 unique LTL-NL pairs from each category, creating a dedicated evaluation set of 4,000 examples which we then used for the evals.

The table above shows that logical complexity is a primary driver of difficulty. The model maintains reasonable performance on formulas with low to medium complexity but struggles to preserve the precise semantic structure of more deeply nested LTL expressions when translating from natural language. This highlights a key area for future work: developing architectures that are more robust to increases in logical complexity.

## B.2 Generalization to unseen domains

We tested the limits of our models in two distinct and ambitious ways: (1) generalization to unseen domains within the same LTL formalism and (2) emergent generalization to an entirely new, unseen formalism (Signal Temporal Logic).

First, to directly assess cross-domain generalization, we performed a new set of experiments using a Leave-One-Domain-Out (LODO) cross-validation methodology. We trained our Llama-3-8B-Instruct (FT) model on 12 of the 13 domains and tested its performance on the held-out domain. The results for three representative held-out domains are presented in Table below. "In-Domain" refers to the original performance when the model was trained on all 13 domains. "Out-of-Domain" is the performance on the domain when it was held out from the training set.

Table 10: In-Domain vs. Out-of-Domain Semantic Equivalence

| Held-Out Domain | In-Domain SemEq (%) | Out-of-Domain SemEq (%) |
|---|---|---|
| Aerospace | 27.5 | 19.2 |
| Home Automation | 30.1 | 22.5 |
| Financial/Transaction Sys. | 25.8 | 16.7 |

The LODO results show an expected decrease in performance when the model encounters a domain it has not been trained on. However, the model retains a significant portion of its capability, achieving semantic equivalence scores between 16.7% and 22.5% in a zero-shot setting. This indicates that the model is not merely memorizing domain-specific patterns but is successfully transferring learned logical structures to new contexts, demonstrating a solid degree of domain generalization.

Second, we conducted an experiment to investigate if the model, fine-tuned only on LTL, could show emergent generalization to a different temporal logic. We tested the same VERIFY-finetuned Llama-3-8B model on a curated benchmark of 100 human-written Signal Temporal Logic (STL) specifications. STL is a related but distinct formalism used for real-valued signals, which the model had never seen. The model was evaluated zero-shot, without any fine-tuning on STL-specific data. The results are shown in the Table below. The human correctness ratings were provided by the paper authors, with a scoring of all 100 examples on a 1-to-5 scale.

Table 11: Performance on Core Translation Tasks

| Task | Metric | Performance |
|---|---|---|
| STL $\rightarrow$ NL Generation | Human-rated Correctness (1–5) | 3.7 / 5.0 |
| NL $\rightarrow$ STL Translation | Semantic Equivalence (%) | 14.3% |

Remarkably, the model demonstrates a non-trivial ability to operate on STL specifications. It can generate coherent and largely correct natural language descriptions from STL formulas and can even parse NL into semantically valid STL with 14.3% accuracy. That the model achieves this capability without any exposure to STL suggests it has learned some of the fundamental, underlying principles of temporal logic that are common to both LTL and STL, rather than just the surface syntax of LTL.

### B.3 COMMON ERROR PATTERNS

It is important to look at the common error patterns in tasks 1-5. To do this, we performed an error analysis on the outputs of the Llama-3-8B model, focusing on the most challenging NL → LTL translation task. The results, based on a manual review of 100 incorrect predictions, are summarized in the Table below. Based on a manual review of 100 incorrect predictions.

Table 12: Error Analysis of NL to LTL Generation

| Error Category | Frequency | Description |
|---|---|---|
| Incorrect Logical Scope | 41% | Model fails to correctly capture operator precedence and scope from the NL sentence, often misplacing parentheses or nesting clauses incorrectly. |
| Temporal Operator Mismatch | 28% | Model confuses semantically close temporal operators, most commonly substituting 'Until' (U) for 'Weak Until' (W) or 'Globally' (G) for 'Finally' (F). |
| Propositional Atom Error | 17% | Model either fails to include a required propositional atom from the context or hallucinates an atom that was not specified. |
| Contextual Grounding Failure | 9% | The generated LTL is logically sound but fails to correctly incorporate the specific variable definitions provided in the 'activity' context. |
| Syntactic Malformation | 5% | The output is not a syntactically valid LTL formula and cannot be parsed. |

Our analysis reveals that outright syntactic errors are rare (5%). Instead, the majority of failures are semantic in nature. The most frequent issue (41%) is the model's struggle to correctly capture the precedence and scope of operators from complex natural language sentences. Furthermore, the model often has difficulty distinguishing between strong and weak temporal requirements (e.g., 'Until' vs. 'Weak Until'), accounting for 28% of errors.

## C  FULL LTL-TO-ITL COMPLETENESS PROOF

This appendix provides a formal argument for the completeness of the mapping from the Linear Temporal Logic (LTL) fragment used in the VERIFY dataset to the canonical Intermediate Technical Language (ITL) representation generated by our pipeline. Completeness, in this context, means that every LTL formula within the defined fragment can be successfully and deterministically translated into a well-defined ITL string.

### C.1  SYNTAX OF THE SOURCE LTL FRAGMENT (LTL$_{VF}$)

The LTL formulas $(\phi, \psi)$ in the VERIFY dataset are generated and subsequently verified to conform to a specific syntactic fragment, denoted LTL$_{VF}$. Let $AP = \{p, q, r, s, t, u, v, w\}$ be the finite set of atomic propositions used in our dataset. The set of well-formed formulas in LTL$_{VF}$ is defined inductively as the smallest set satisfying the following rules:

1. **Atomic Proposition:** If $\alpha \in AP$, then $\alpha \in$ LTL$_{VF}$.

2. **Boolean Constants:** $\top$ (true) $\in$ LTL$_{VF}$ and $\bot$ (false) $\in$ LTL$_{VF}$.

3. **Negation:** If $\phi \in$ LTL$_{VF}$, then $\neg\phi \in$ LTL$_{VF}$.

4. **Conjunction:** If $\phi, \psi \in$ LTL$_{VF}$, then $(\phi \wedge \psi) \in$ LTL$_{VF}$.

5. **Disjunction:** If $\phi, \psi \in$ LTL$_{VF}$, then $(\phi \vee \psi) \in$ LTL$_{VF}$.

6. **Implication:** If $\phi, \psi \in$ LTL$_{VF}$, then $(\phi \rightarrow \psi) \in$ LTL$_{VF}$.

7. **Equivalence:** If $\phi, \psi \in$ LTL$_{VF}$, then $(\phi \leftrightarrow \psi) \in$ LTL$_{VF}$.

8. **Next:** If $\phi \in$ LTL$_{VF}$, then $X\phi \in$ LTL$_{VF}$.

9. **Globally (Always):** If $\phi \in$ LTL$_{VF}$, then $G\phi \in$ LTL$_{VF}$.

10. **Finally (Eventually):** If $\phi \in \text{LTL}_{\text{VF}}$, then $F\phi \in \text{LTL}_{\text{VF}}$.

11. **Until:** If $\phi, \psi \in \text{LTL}_{\text{VF}}$, then $(\phi\, U\, \psi) \in \text{LTL}_{\text{VF}}$.

12. **Release:** If $\phi, \psi \in \text{LTL}_{\text{VF}}$, then $(\phi\, R\, \psi) \in \text{LTL}_{\text{VF}}$.

13. **Weak Until:** If $\phi, \psi \in \text{LTL}_{\text{VF}}$, then $(\phi\, W\, \psi) \in \text{LTL}_{\text{VF}}$.

14. **Strong Release (Matches):** If $\phi, \psi \in \text{LTL}_{\text{VF}}$, then $(\phi\, M\, \psi) \in \text{LTL}_{\text{VF}}$.

Standard operator precedence and parentheses are used for disambiguation. All LTL formulas included in VERIFY are parsed and canonicalized by the Spot library (version 2.11.6), ensuring they conform to this fragment and have a standardized representation. We assume the standard semantics of LTL over infinite traces (Pnueli, 1977; Clarke et al., 2018).

## C.2 STRUCTURE OF THE CANONICAL INTERMEDIATE TECHNICAL LANGUAGE (ITL$_{\text{CANONICAL}}$)

The canonical ITL (ITL$_{\text{Canonical}}$) is not defined by an independent generative grammar but is rather procedurally generated from the AST of an LTL formula. It results in structured English strings composed of atomic proposition identifiers, specific keywords/phrases corresponding to LTL operators, and punctuation (primarily commas and parentheses that mirror the LTL structure). The core keywords and templates for ITL$_{\text{Canonical}}$ (derived from the mapping rules discussed in Section 4.2) are defined as follows (where $\phi'_{ITL}$ and $\psi'_{ITL}$ represent the ITL translations of LTL subformulas $\phi$ and $\psi$ respectively):

- Atomic proposition $\alpha$: maps to its string representation (e.g., "$p$").
- $\top$: maps to "true".
- $\bot$: maps to "false".
- $\neg\phi$: maps to "not $\phi'_{ITL}$".
- $\phi \wedge \psi$: maps to "$\phi'_{ITL}$ and $\psi'_{ITL}$".
- $\phi \vee \psi$: maps to "$\phi'_{ITL}$ or $\psi'_{ITL}$".
- $\phi \rightarrow \psi$: maps to "if $\phi'_{ITL}$, then $\psi'_{ITL}$".
- $\phi \leftrightarrow \psi$: maps to "$\phi'_{ITL}$ if and only if $\psi'_{ITL}$".
- $X\phi$: maps to "In the next state, $\phi'_{ITL}$".
- $G\phi$: maps to "Always, $\phi'_{ITL}$".
- $F\phi$: maps to "Eventually, $\phi'_{ITL}$".
- $\phi\, U\, \psi$: maps to "$\phi'_{ITL}$ until $\psi'_{ITL}$".
- $\phi\, R\, \psi$: maps to "$\phi'_{ITL}$ releases $\psi'_{ITL}$".
- $\phi\, W\, \psi$: maps to "$\phi'_{ITL}$ weakly until $\psi'_{ITL}$".

The generation process ensures that the nesting and scope of operators in the LTL formula are preserved in the hierarchical structure implied by the ITL string composition, often through implicit parenthesization mirroring the LTL AST structure.

## C.3 FORMAL DEFINITION OF THE MAPPING FUNCTION $\mathcal{T}$

We define the mapping function $\mathcal{T} : \text{LTL}_{\text{VF}} \rightarrow \text{Strings}$, which translates an LTL formula $\phi \in \text{LTL}_{\text{VF}}$ (assumed to be in its Spot-canonical form and represented as an AST, denoted $\text{AST}(\phi)$) into its ITL$_{\text{Canonical}}$ string representation. The function $\mathcal{T}$ is defined recursively based on the structure of $\text{AST}(\phi)$:

1. If $\phi = \alpha$ where $\alpha \in AP$: $\mathcal{T}(\text{AST}(\alpha)) = \text{``}\alpha\text{''}$ (the string literal of the atom).
2. If $\phi = \top$: $\mathcal{T}(\text{AST}(\top)) = \text{``true''}$.
3. If $\phi = \bot$: $\mathcal{T}(\text{AST}(\bot)) = \text{``false''}$.
4. If $\phi = \neg\psi$: $\mathcal{T}(\text{AST}(\neg\psi)) = \text{``not ''} \oplus \mathcal{T}(\text{AST}(\psi))$, where $\oplus$ denotes string concatenation.
5. If $\phi = \psi_1 \wedge \psi_2$: $\mathcal{T}(\text{AST}(\psi_1 \wedge \psi_2)) = \mathcal{T}(\text{AST}(\psi_1)) \oplus \text{`` and ''} \oplus \mathcal{T}(\text{AST}(\psi_2))$.

6. If $\phi = \psi_1 \vee \psi_2$: $\mathcal{T}(\mathrm{AST}(\psi_1 \vee \psi_2)) = \mathcal{T}(\mathrm{AST}(\psi_1)) \oplus$ " or " $\oplus \mathcal{T}(\mathrm{AST}(\psi_2))$.

7. If $\phi = \psi_1 \rightarrow \psi_2$: $\mathcal{T}(\mathrm{AST}(\psi_1 \rightarrow \psi_2)) =$ "if " $\oplus \mathcal{T}(\mathrm{AST}(\psi_1)) \oplus$ ", then " $\oplus \mathcal{T}(\mathrm{AST}(\psi_2))$.

8. If $\phi = \psi_1 \leftrightarrow \psi_2$: $\mathcal{T}(\mathrm{AST}(\psi_1 \leftrightarrow \psi_2)) = \mathcal{T}(\mathrm{AST}(\psi_1)) \oplus$ " if and only if " $\oplus \mathcal{T}(\mathrm{AST}(\psi_2))$.

9. If $\phi = X\psi$: $\mathcal{T}(\mathrm{AST}(X\psi)) =$ "In the next state, " $\oplus \mathcal{T}(\mathrm{AST}(\psi))$.

10. If $\phi = G\psi$: $\mathcal{T}(\mathrm{AST}(G\psi)) =$ "Always, " $\oplus \mathcal{T}(\mathrm{AST}(\psi))$.

11. If $\phi = F\psi$: $\mathcal{T}(\mathrm{AST}(F\psi)) =$ "Eventually, " $\oplus \mathcal{T}(\mathrm{AST}(\psi))$.

12. If $\phi = \psi_1 \, U \, \psi_2$: $\mathcal{T}(\mathrm{AST}(\psi_1 \, U \, \psi_2)) = \mathcal{T}(\mathrm{AST}(\psi_1)) \oplus$ " until " $\oplus \mathcal{T}(\mathrm{AST}(\psi_2))$.

13. If $\phi = \psi_1 \, R \, \psi_2$: $\mathcal{T}(\mathrm{AST}(\psi_1 \, R \, \psi_2)) = \mathcal{T}(\mathrm{AST}(\psi_1)) \oplus$ " releases " $\oplus \mathcal{T}(\mathrm{AST}(\psi_2))$.

14. If $\phi = \psi_1 \, W \, \psi_2$: $\mathcal{T}(\mathrm{AST}(\psi_1 \, W \, \psi_2)) = \mathcal{T}(\mathrm{AST}(\psi_1)) \oplus$ " weakly until " $\oplus \mathcal{T}(\mathrm{AST}(\psi_2))$.

15. If $\phi = \psi_1 \, M \, \psi_2$: $\mathcal{T}(\mathrm{AST}(\psi_1 \, M \, \psi_2)) = \mathcal{T}(\mathrm{AST}(\psi_1)) \oplus$ " strong release " $\oplus \mathcal{T}(\mathrm{AST}(\psi_2))$.

Parentheses in the output ITL string are implicitly handled by the recursive structure of $\mathcal{T}$ and the string concatenations, preserving the LTL AST's operator scope and precedence. Explicit parentheses can be added around the ITL for sub-formulas in practice to ensure clarity, especially for binary operators, e.g., $\mathcal{T}(\mathrm{AST}(\psi_1 \wedge \psi_2)) =$ "(" $\oplus \mathcal{T}(\mathrm{AST}(\psi_1)) \oplus$ ') and (' $\oplus \mathcal{T}(\mathrm{AST}(\psi_2)) \oplus$ ')". However, for this proof, the direct template application is sufficient as structure is inherited from the AST.

### C.4 PROOF OF COMPLETENESS (TOTALITY OF $\mathcal{T}$)

We claim that the mapping function $\mathcal{T}$ is total for all LTL formulas $\phi \in \mathrm{LTL_{VF}}$. That is, for every valid LTL formula generated and verified in our dataset (which conforms to $\mathrm{LTL_{VF}}$), $\mathcal{T}$ produces a well-defined $\mathrm{ITL_{Canonical}}$ string output. The proof proceeds by structural induction on the formula $\phi$.

**Base Cases:**

- If $\phi = \alpha$, where $\alpha \in AP$: $\mathcal{T}(\mathrm{AST}(\alpha))$ is defined as the string literal "$\alpha$".

- If $\phi = \top$: $\mathcal{T}(\mathrm{AST}(\top))$ is defined as the string "true".

- If $\phi = \bot$: $\mathcal{T}(\mathrm{AST}(\bot))$ is defined as the string "false".

In all base cases, $\mathcal{T}$ yields a well-defined string.

**Inductive Hypothesis (IH):** Assume that for any LTL formula $\psi$ (and $\chi$, if applicable) that is a proper subformula of $\phi$, the function $\mathcal{T}$ is total, and $\mathcal{T}(\mathrm{AST}(\psi))$ (and $\mathcal{T}(\mathrm{AST}(\chi))$) is a well-defined ITL string.

**Inductive Step:** We examine each case for constructing $\phi$ from its subformula(s) according to the rules of $\mathrm{LTL_{VF}}$:

1. If $\phi = \neg\psi$: By the IH, $\mathcal{T}(\mathrm{AST}(\psi))$ is a well-defined string. The rule for $\neg$ (Rule 4 in the definition of $\mathcal{T}$) defines $\mathcal{T}(\mathrm{AST}(\neg\psi))$ as the concatenation "not " $\oplus \mathcal{T}(\mathrm{AST}(\psi))$. This operation on well-defined strings results in a well-defined string.

2. If $\phi = \psi_1 \, \mathrm{op} \, \psi_2$, where $\mathrm{op} \in \{\wedge, \vee, \rightarrow, \leftrightarrow, U, R, W\}$: By the IH, $\mathcal{T}(\mathrm{AST}(\psi_1))$ and $\mathcal{T}(\mathrm{AST}(\psi_2))$ are well-defined strings. The rules for these binary operators (Rules 5-8, 12-14 in the definition of $\mathcal{T}$) define $\mathcal{T}(\mathrm{AST}(\phi))$ as a concatenation of $\mathcal{T}(\mathrm{AST}(\psi_1))$, a specific ITL keyword for op, and $\mathcal{T}(\mathrm{AST}(\psi_2))$. This results in a well-defined string.

3. If $\phi = \mathrm{op} \, \psi$, where $\mathrm{op} \in \{X, G, F\}$: By the IH, $\mathcal{T}(\mathrm{AST}(\psi))$ is a well-defined string. The rules for these unary temporal operators (Rules 9-11 in the definition of $\mathcal{T}$) define $\mathcal{T}(\mathrm{AST}(\phi))$ as a concatenation of the ITL keyword for op and $\mathcal{T}(\mathrm{AST}(\psi))$. This results in a well-defined string.

Since the base cases hold and the inductive step covers all LTL operators defined in $\mathrm{LTL_{VF}}$, the function $\mathcal{T}$ is total for all formulas $\phi \in \mathrm{LTL_{VF}}$.

## C.5 Preservation of Semantic Structure and Reversibility

**Semantic Structure Preservation:** The function $\mathcal{T}$ is designed to be structure-preserving. It operates directly on the AST derived from the Spot-parsed (and canonicalized) LTL formula. The recursive definition of $\mathcal{T}$ ensures a one-to-one mapping between LTL operators in the AST and their corresponding ITL keywords/templates. The recursive application of these mappings ensures that the nesting and scope of operators in the LTL formula are preserved in the hierarchical structure of the resulting $ITL_{Canonical}$ string. This structural isomorphism provides a strong basis for asserting that the core semantic relationships (temporal and logical) of the LTL formula are maintained in its ITL translation. A formal proof of semantic equivalence would require a formal semantics for ITL; however, the systematic, structure-driven nature of $\mathcal{T}$ supports this claim.

**Reversibility (ITL to LTL):** The $ITL_{Canonical}$ strings generated by $\mathcal{T}$ are designed to be unambiguously parsable back into LTL formulas that are semantically equivalent to the original LTL formulas. This reversibility is crucial for verifying the integrity of the ITL representation. As described in Section 4.2, an ITL-to-LTL parser was developed based on the inverse of the 'LTL_TO_CANONICAL' rules. The VERIFY dataset construction pipeline includes an automated verification step where canonical ITL strings are parsed back to LTL, and this reconstructed LTL is then formally checked for semantic equivalence against the original Spot-verified LTL formula using Spot's built-in capabilities (e.g., 'spot.are_equivalent()'). This empirical validation across the dataset (specifically, 18% of it, as mentioned in Section 4.4 for NL, and a similar process for ITL integrity check mentioned in Section 4.2) confirms that the LTL $\rightarrow$ ITL $\rightarrow$ LTL round trip preserves logical meaning.

## C.6 Conclusion

The mapping function $\mathcal{T}$ from the defined and verified LTL fragment $LTL_{VF}$ to $ITL_{Canonical}$ is total (complete), meaning every formula in $LTL_{VF}$ has a corresponding ITL string. This mapping is deterministic and preserves the structural composition of the LTL formula. Empirical verification through round-trip LTL-ITL-LTL conversion and semantic equivalence checking using formal tools (Spot) further confirms that the generated canonical ITL accurately represents the logical meaning of the source LTL formula. Therefore, the grammar used for translating LTL to $ITL_{Canonical}$ is complete with respect to the $LTL_{VF}$ fragment.

# D Extended Dataset Details

This appendix provides further details about the VERIFY dataset, including additional illustrative examples, comprehensive per-domain statistics, supplementary visualizations, and the full data schema.

## D.1 Additional Examples

To further illustrate the nature and diversity of the VERIFY dataset, this section presents five detailed examples. Each example includes the domain, the natural language definitions for propositional variables (Activity), the Spot-canonical Linear Temporal Logic (LTL) formula, its corresponding rule-based canonical Intermediate Technical Language (ITL) representation, and the final contextual Natural Language (NL) translation. These examples showcase variations in logical complexity, domain-specific terminology, and the types of properties represented.

### D.1.1 Example 1: Automotive/Autonomous Vehicles

- **Domain:** Automotive/Autonomous Vehicles
- **Activity:**
    - `p`: Lane departure detected
    - `q`: Obstacle detection active
    - `r`: Driver override requested
    - `s`: Sensor calibration complete
    - `t`: Emergency braking engaged

- – u: System in autonomous mode
- – v: GPS signal lost
- – w: Manual steering input detected

- **LTL Formula (Spot-canonical):**

$$(X\neg(qRp \rightarrow Xs)\,U\,(p \rightarrow q)\,RXu \rightarrow \neg X(q \vee q)\,R\,(p \rightarrow s)U\,(pUwWr \rightarrow (r \wedge q \leftrightarrow Gq)))$$
$$U\,((Fq \vee Xr)\,R\,(\neg s)\,WrRuU\,(q \wedge (v \vee v) \leftrightarrow (r \vee t)\,Wq) \leftrightarrow$$
$$(GrWq \rightarrow (t \rightarrow w)\,Wu \rightarrow (vU\,(u \leftrightarrow r) \leftrightarrow t))\,W\,(\neg Gv \vee p))$$
$$\vee\,(t \rightarrow ((t \vee q) \wedge Fp)\,WrWrRu)\,RrUrR\,(\neg s)$$

- **Canonical ITL:**
  if In the next state, not if q releases p, then In the next state, s until if p, then q releases In the next state, u, then not In the next state, q or q releases if p, then s until if p until w weakly until r, then q and r if and only if Always, q until Eventually, q or In the next state, r releases not s weakly until r releases u until q and v or v if and only if r or t weakly until q if and only if if Always, r weakly until q, then if if t, then w weakly until u, then t if and only if v until r if and only if u weakly until p or not Always, v or if t, then q or t and Eventually, p weakly until r weakly until r releases u releases r until r releases not s

- **NL Translation:** The system must maintain that after lane departure, either obstacle detection remains active until sensor calibration follows, or autonomous mode persists until manual override triggers a protocol where persistent lane-keeping requires continuous obstacle detection, until either emergency braking initiates redundant navigation checks or GPS loss forces driver takeover – whichever occurs first while maintaining sensor integrity throughout.

### D.1.2 EXAMPLE 2: HOME AUTOMATION

- **Domain:** Home Automation

- **Activity:** Atomic propositions represent device states: u=user presence detected, r=security system armed, q=lights activated, s=door locked, t=motion detected, v=HVAC running, w=window open, p=power saving mode

- **LTL Formula (Spot-canonical):**

$$GGuW\,(r \rightarrow q) \rightarrow qU\,(\neg r)W\,((s \leftrightarrow r) \rightarrow r)$$
$$U\,(s \wedge XF(t \rightarrow q)\,UrWvRX(q \vee r)\,WFwW\,((\neg s)\,WXw \wedge (v \vee u \rightarrow u)))$$
$$\vee\,((v \wedge \neg p)\,U\,(r \rightarrow w)\,WrRu \rightarrow (Gs \vee w)\,Wv)\,W\,(X(v \rightarrow t)\,RFu \wedge v)$$
$$U\,((\neg s \vee Fv)\,WFs \wedge qRu \leftrightarrow Xv)\,Uq$$

- **Canonical ITL:**
  if Always, u weakly until if r, then q, then q until not r weakly until if r if and only if s, then r until s and In the next state, Eventually, if t, then q until r weakly until v releases In the next state, q or r weakly until Eventually, w weakly until if u or v, then u and not s weakly until In the next state, w or if not p and v until if r, then w weakly until r releases u, then w or Always, s weakly until v weakly until v and In the next state, if v, then t releases Eventually, u until not s or Eventually, v weakly until Eventually, s and q releases u if and only if In the next state, v until q

- **NL Translation:** If user presence persists until security arming triggers lights, then lights must stay active until disarm occurs, maintaining that locked doors require security engagement until motion triggers lights through HVAC operation, while windows stay closed unless presence or HVAC override occurs, and power mode exits when security triggers window control until HVAC/temperature conditions maintain either sustained locking or window states matching security status through presence verification.

### D.1.3 EXAMPLE 3: SECURITY AND AUTHENTICATION

- **Domain:** Security and Authentication

- **Activity:**
  - p: Unauthorized access detected
  - q: User authentication request
  - r: System lockdown activated
  - s: Security breach detected
  - t: Two-factor authentication enabled
  - u: User role changed
  - v: Vulnerability patched
  - w: Access granted

- **LTL Formula (Spot-canonical):**

$$((\neg((t \leftrightarrow q) \vee s) R (\neg p) RFp \vee (\neg t \vee s) WGrU (XGr \to t)) U$$
$$(\neg s) W (r \leftrightarrow Fq) WvWv \leftrightarrow v$$
$$W ((\neg\neg s) Wu \wedge (p \vee FpUpR (s \to s) \to (w \leftrightarrow w))$$
$$W (\neg(Gq \vee u) U (\neg s \vee r) U (q \leftrightarrow t)))$$

- **Canonical ITL:**
  v if and only if not s or q if and only if t releases not p releases Eventually, p or s or not t weakly until Always, r until if In the next state, Always, r, then t until not s weakly until r if and only if Eventually, q weakly until v weakly until s weakly until u

- **NL Translation:** The system must maintain that: either (1) security breach absence and two-factor-authentication alignment with authentication requests persist while blocking unauthorized access until detection occurs, or (2) lockdown persists until two-factor activation if future lockdown continuity implies it, all until breaches cease. This continues weakly until breaches force lockdown equivalence to pending authentications, repeated vulnerability patching, and role changes persist only if access-right consistency holds unless infinite authentication demands or role changes occur until security states resolve.

### D.1.4 EXAMPLE 4: SMART GRID/ENERGY MANAGEMENT

- **Domain:** Smart Grid/Energy Management
- **Activity:** p=peak load condition, t=tariff adjustment activated, r=renewable generation available, u=usage restriction enforced, w=wind power input threshold, q=grid stability query issued, s=storage system activated, v=voltage stability compromised

- **LTL Formula (Spot-canonical):**

$$(pWFXp \leftrightarrow t)$$
$$R ((rUp \vee (t \to t) \wedge \neg r) U (uWr \wedge XpUw) \wedge ((q \wedge p \to s) \wedge (q \to Fq) \to (s \to s))$$
$$W (t \wedge u) W (uWv \to Xt) \leftrightarrow \neg Fr \to vR (\neg w) R (\neg v \leftrightarrow t))$$
$$U (vR ((u \vee rRw \wedge Fv) \wedge Gs \leftrightarrow GrUpWuWFs) \leftrightarrow XX(\neg s) U (s \leftrightarrow q)U$$
$$Fu \vee (s \to (q \to r) Uu) \wedge (\neg q \leftrightarrow v) Ru$$

- **Canonical ITL:**
  t if and only if p weakly until Eventually, In the next state, p releases not r or r until p until u weakly until r and In the next state, p until w if and only if if not Eventually, r, then v releases not w releases t if and only if not v until v releases u or r releases w and Eventually, v and Always, s if and only if Always, r until p weakly until u weakly until Eventually, s if and only if In the next state, In the next state, not s until q if and only if s until Eventually, u or if s, then if q, then r until u and not q if and only if v releases u

- **NL Translation:** Tariff adjustments match peak load persistence until eventual resumption if and only if grid operations maintain: renewable availability until peak load or usage restrictions with wind thresholds, requiring storage activation only when stability queries trigger sustained responses, unless voltage instability forces delayed demand response until tariff-voltage alignment governs restoration.

Table 13: Application domains covered in the VERIFY dataset.

| Domain | Illustrative Context Example Snippet (activity) |
|---|---|
| Financial Services | p=trade execution confirmed, q=risk limit check passed |
| Web Services / E-commerce | p=user adds item to cart, q=inventory level updated |
| Home Automation | p=motion detected in room, q=lights turn on |
| Aerospace / Avionics | p=altitude within safe range, q=autopilot engaged |
| Medical Devices | p=heart rate exceeds threshold, q=alert generated |
| Industrial Automation / Mfg. | p=pressure threshold reached, q=safety valve opens |
| Automotive Systems | p=obstacle detected by sensor, q=emergency brake applied |
| Robotics / Autonomous Systems | p=battery level low, q=robot returns to charging station |
| Network Protocols / Security | p=login attempt failed 3 times, q=account locked |
| Business Process Management | p=invoice approved, q=payment scheduled |
| Supply Chain / Logistics | p=package scanned at hub, q=tracking status updated |
| Energy Systems / Smart Grid | p=demand exceeds supply, q=load shedding initiated |
| Telecommunications | p=call successfully connected, q=billing record created |

### D.1.5 EXAMPLE 5: VERSION CONTROL AND CODE REVIEWS

- **Domain:** Version Control and Code Reviews
- **Activity:** `q`: Code review requested | `v`: Code review passed | `u`: Code conflicts resolved | `t`: Tests passed | `w`: Work-in-progress flag | `p`: Pull request open | `r`: Revision submitted | `s`: Code merged
- **LTL Formula (Spot-canonical):**

$$qU\left(v \vee \left(\left(\left(u \to t\right) \wedge \left(w \vee p\right)\right) R\left(\neg t\right) U\left(p \vee r\right) \leftrightarrow \neg r\right) U\left(p \to \left(Fw \to Xs\right) Ws\right)\right)$$
$$U\left(\left(vU\left(t \wedge v\right) Rs \to \neg XtWwRtRp \vee Gq\right) \wedge \left(s \leftrightarrow FwUrW\left(w \leftrightarrow u\right)\right) WGFtWGs\right.$$
$$\left.W\left(\left(t \wedge u\right) Rp \vee w\right) W\left(GsUwUqRs \vee \left(\left(Xt \to \left(r \leftrightarrow r\right)\right) \wedge \left(\neg p\right) UuRt \leftrightarrow w\right)\right)\right)$$

- **Canonical ITL:**
  q until v or not r if and only if if u, then t and p or w releases not t until p or r until if p, then if Eventually, w, then In the next state, s weakly until s until if v until t and v releases s, then not In the next state, t weakly until w releases t releases p or Always, q and s if and only if Eventually, w until r weakly until u if and only if w weakly until Always, Eventually, t weakly until Always, s weakly until w or t and u releases p weakly until Always, s until w until q releases s or w if and only if not p until u releases t

- **NL Translation:** A code review remains requested until either it passes, or (if conflicts being resolved guarantees tests pass and an open pull request or WIP flag persists while tests are failing until a revision or pull request exists) exactly when no revision exists, until pull requests being open implies (if work eventually continues, the next state must merge code *or* keep merging pending) persists, while either: (1) review passes until tests succeed with passing review under merge protection until tests require WIP or persistent review requests; or (2) merging occurs only if eventual WIP under revision constraints matches conflict resolution, weakly until recurring tests and merges align with open pull requests or WIP, provided merges persist until WIP transitions or review compliance.

### D.2 PER-DOMAIN STATISTICS

To provide a deeper insight into the characteristics of the VERIFY dataset across its 13 domains, Table 13 summarizes the application domains covered VERIFY and Table 14 summarizes key statistics. These include the number of unique LTL formulas, various measures of LTL formula complexity (average, median, min/max number of temporal operators, and AST depth), natural language translation length statistics (word count), activity string length statistics (word count), and approximate vocabulary sizes for both translations and activities within each domain.

### D.2.1 LTL OPERATOR AND SUB-PATTERN FREQUENCIES PER DOMAIN

The distribution of LTL operators and common structural patterns (identified by Spot's formula kinds) varies across domains, reflecting different specification needs. Below is a summary of the frequency

Table 14: Detailed Per-Domain Dataset Statistics. "LTL Ops" refers to the count of temporal operators. "LTL Depth" refers to the AST depth. "Words" refers to word count. "Vocab Size" is the count of unique words (lowercase, simple tokenization).

| Domain | Unique LTLs | LTL Operators | | | LTL Depth | | NL Trans. Words | | | Activity Words | | | Vocab Size | |
|---|---|---|---|---|---|---|---|---|---|---|---|---|---|---|
| | | Avg | Med | Min/Max | Avg | Med | Avg | Med | Min/Max | Avg | Med | Min/Max | NL | Activity |
| Aerospace | 16821 | 6.0 | 6 | 0/19 | 4.98 | 5 | 56.21 | 56 | 10/152 | 33.99 | 34 | 6/91 | ~4034 | ~5810 |
| Auto/Autonomous | 16711 | 6.0 | 6 | 0/22 | 5.04 | 5 | 56.17 | 56 | 10/162 | 29.83 | 30 | 5/88 | ~3749 | ~6318 |
| Build Pipelines/CI-CD | 16737 | 6.0 | 6 | 0/21 | 4.98 | 5 | 53.60 | 54 | 6/152 | 24.50 | 24 | 5/85 | ~2975 | ~3525 |
| Financial/Transaction | 16765 | 6.0 | 6 | 0/18 | 4.98 | 5 | 53.26 | 53 | 7/140 | 26.54 | 26 | 5/130 | ~3469 | ~3507 |
| Home Automation | 16748 | 6.0 | 6 | 0/18 | 5.01 | 5 | 56.82 | 57 | 7/142 | 32.31 | 32 | 6/74 | ~2961 | ~3019 |
| Industrial Automation | 16782 | 6.2 | 6 | 0/22 | 5.17 | 5 | 54.70 | 55 | 5/201 | 28.27 | 28 | 5/76 | ~3943 | ~4934 |
| Medical Devices | 16710 | 6.0 | 6 | 0/19 | 5.01 | 5 | 56.04 | 56 | 6/144 | 34.86 | 35 | 6/93 | ~4045 | ~4988 |
| Networking/Distributed | 16748 | 5.9 | 6 | 0/21 | 4.96 | 5 | 52.75 | 53 | 11/142 | 28.08 | 28 | 6/78 | ~3636 | ~3771 |
| Robotics | 16800 | 6.0 | 6 | 0/19 | 4.98 | 5 | 56.66 | 57 | 5/204 | 33.71 | 34 | 6/78 | ~3854 | ~4537 |
| Security/Authentication | 16750 | 6.0 | 6 | 0/18 | 4.98 | 5 | 54.31 | 54 | 8/162 | 31.14 | 31 | 6/70 | ~3088 | ~2808 |
| Smart Grid/Energy | 16715 | 6.0 | 6 | 0/21 | 4.98 | 5 | 55.29 | 55 | 8/152 | 32.59 | 32 | 5/78 | ~3395 | ~4039 |
| Version Control | 16820 | 5.9 | 6 | 0/21 | 4.97 | 5 | 55.00 | 55 | 6/185 | 32.98 | 33 | 6/107 | ~3363 | ~3639 |
| Web Services/APIs | 16764 | 5.9 | 6 | 0/19 | 4.96 | 5 | 53.38 | 53 | 6/126 | 28.18 | 28 | 6/98 | ~3691 | ~3675 |
| **Overall Dataset** | ≈15,900* | 5.92 | 5 | 0/44 | 4.99 | 5 | 54.93 | 54 | 5/204 | 31.02 | 31 | 5/130 | ≈18K* | ≈10K* |

\* Total unique LTLs / total unique vocabulary across all domains.
LTL complexity statistics (Ops and Depth) are based on the LTL formulas associated with translations in each domain.
The LTL Operator counts in this table refer to all operators (temporal and boolean), whereas Figure 1a focuses on temporal operators only.

of top-level LTL operators (G, F, X) and common Spot formula kinds (e.g., Implies, U, R, W, Equiv, And, Or) for each domain. This data is derived from analyzing the LTL formulas associated with the NL translations in each respective domain.

- **Aerospace:** Predominantly features 'G' (Global), 'F' (Finally), and 'X' (Next) as top-level operators. Common structural patterns include Implications, Until, and Release. *(Example counts: G: 2059, F: 2055, X: 2055; Implies: 2900, R: 2510, U: 2496)*

- **Automotive/Autonomous Vehicles:** High use of 'X', 'G', and 'F'. Implications, Until, and Release are common patterns. *(Example counts: X: 2119, G: 2058, F: 2051; Implies: 2886, U: 2552, R: 2468)*

- **Build Pipelines and CI/CD:** 'F', 'X', and 'G' are frequent. Structural patterns show many Implications, Until, and Release forms. *(Example counts: F: 2125, X: 2063, G: 1950; Implies: 2865, U: 2529, R: 2481)*

- **Financial/Transaction Systems:** 'X', 'F', 'G' are common. Implications, Release, and Until patterns are prominent. *(Example counts: X: 2118, F: 2069, G: 1998; Implies: 2925, R: 2508, U: 2488)*

- **Home Automation:** Balanced use of 'X', 'F', 'G'. Implications, Release, and Until are frequent structures. *(Example counts: X: 2098, F: 2088, G: 2069; Implies: 2966, R: 2523, U: 2479)*

- **Industrial Automation/Manufacturing:** 'F', 'G', 'X' are prevalent. Implications, Until, and Release patterns are common. *(Example counts: F: 2050, G: 2031, X: 2027; Implies: 2852, U: 2584, R: 2472)*

- **Medical Devices:** 'F', 'X', 'G' appear often. Implications, Weak Until (W), and Release are frequent. *(Example counts: F: 2104, X: 2047, G: 2037; Implies: 2860, W: 2527, R: 2497)*

- **Networking/Distributed Systems:** 'G', 'X', 'F' are common. Implications, Weak Until (W), and Release structures are frequent. *(Example counts: G: 2103, X: 2039, F: 2034; Implies: 2948, W: 2470, R: 2459)*

- **Robotics:** High frequency of 'G', 'F', 'X'. Implications, Until, and Release are common patterns. *(Example counts: G: 2088, F: 2082, X: 2063; Implies: 2958, U: 2479, R: 2464)*

- **Security and Authentication:** 'X', 'F', 'G' are prominent. Structural patterns often involve Implications, Until, and Weak Until (W). *(Example counts: X: 2107, F: 2077, G: 2059; Implies: 2967, U: 2502, W: 2442)*

- **Smart Grid/Energy Management:** 'G', 'F', 'X' are frequent. Implications, Release, and Until patterns are common. *(Example counts: G: 2111, F: 2100, X: 2040; Implies: 2938, R: 2509, U: 2432)*

- **Version Control and Code Reviews:** 'G', 'F', 'X' appear often. Implications, Weak Until (W), and Until structures are frequently used. *(Example counts: G: 2130, F: 2077, X: 2066; Implies: 2903, W: 2527, U: 2492)*

- **Web Services/APIs:** 'X', 'F', 'G' are common. Implications, Until, and Release are frequent patterns. *(Example counts: X: 2138, F: 2102, G: 2052; Implies: 3020, U: 2557, R: 2501)*

*Note: The operator counts for G, F, X above refer to their appearance as the outermost temporal operator in many formulas within the domain, indicating common high-level properties like invariants, eventualities, or next-state transitions. The structural pattern counts (Implies, U, R, etc.) are derived from Spot's analysis of formula kinds within the LTL expressions for each domain.*

### D.3 ADDITIONAL VISUALIZATIONS

This section presents additional visualizations to complement the main paper, providing further insights into the VERIFY dataset's characteristics.

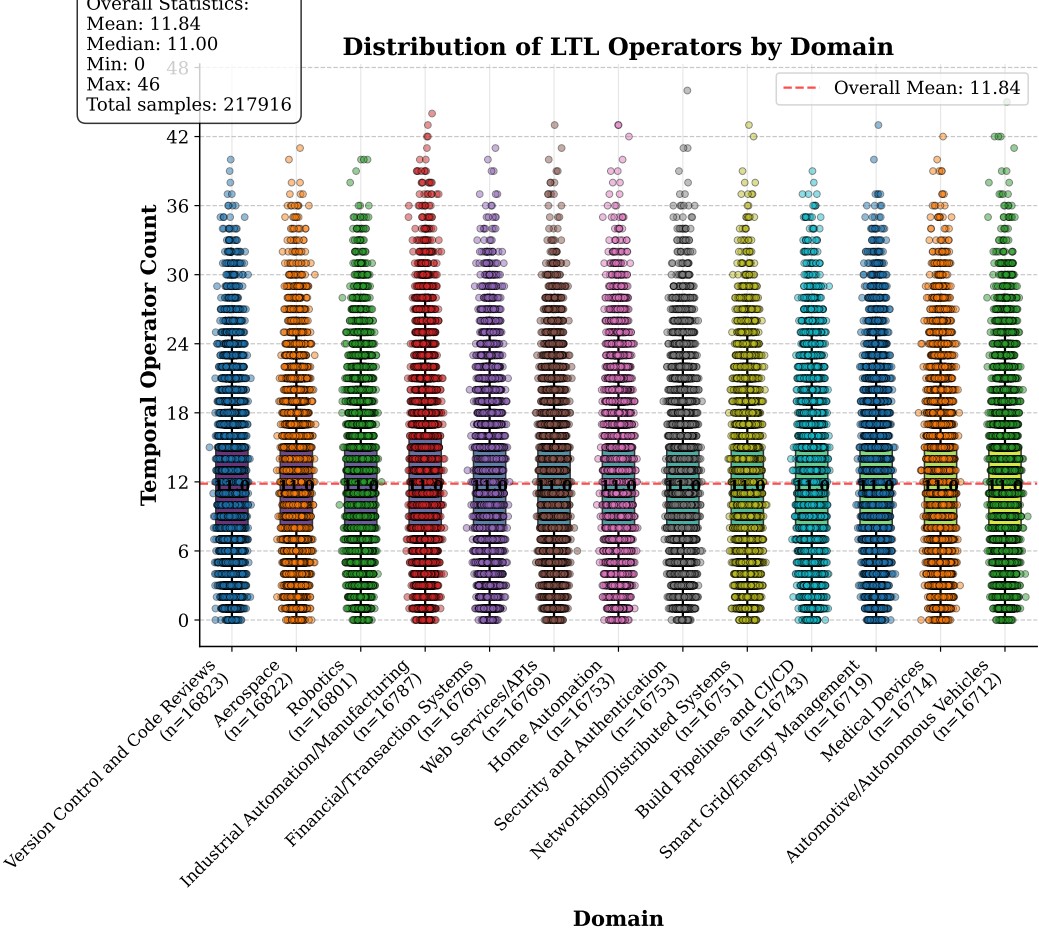

Figure 3: Distribution of LTL temporal operator counts per formula, shown as box plots for each of the 13 domains in the VERIFY dataset. The overall mean is indicated. This complements Figure 1a in the main paper by providing domain-specific views.

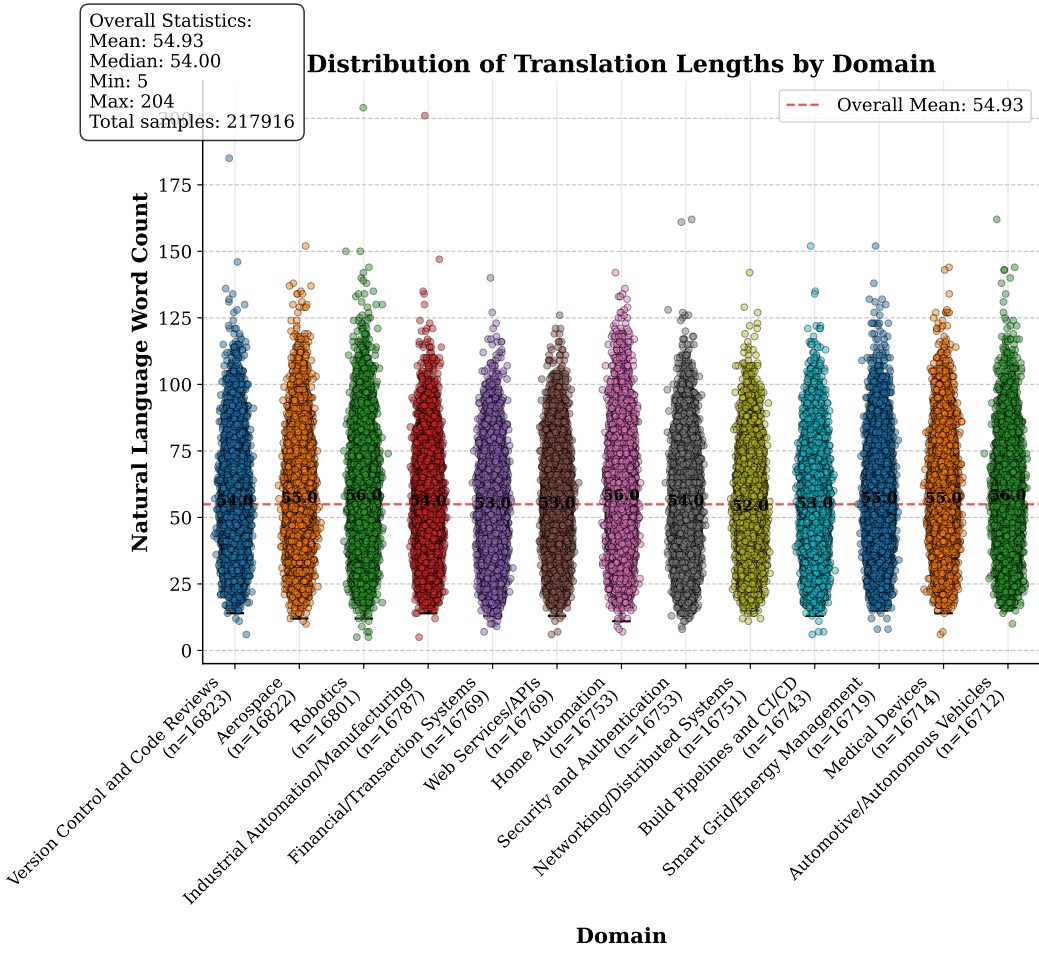

Figure 4: Distribution of natural language translation lengths (by word count) per formula, shown as box plots for each of the 13 domains. The overall mean is indicated. This complements Figure 1b in the main paper with domain-specific distributions.

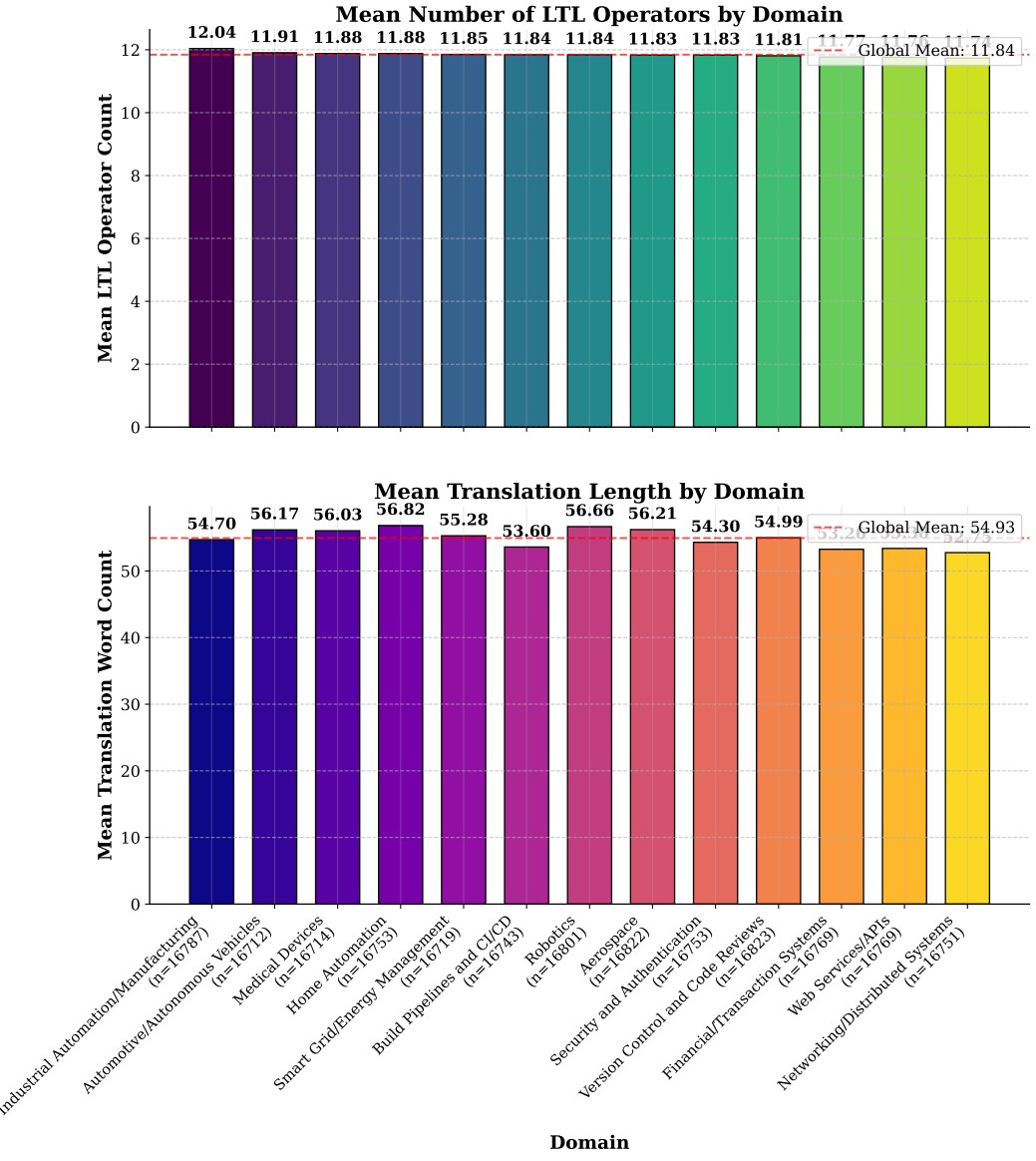

Figure 5: Summary of mean LTL operator counts (top) and mean natural language translation word counts (bottom) across all 13 domains. Global means are indicated by dashed lines. This provides a direct comparison of averages across domains.

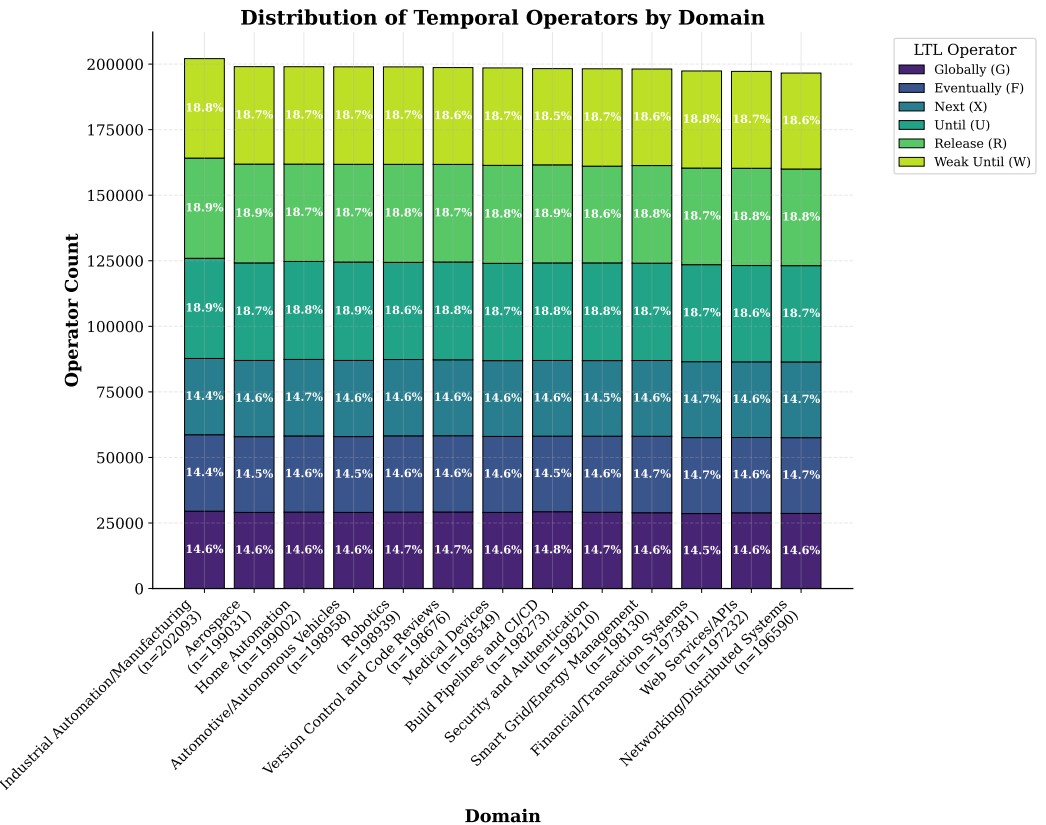

Figure 6: Stacked bar chart illustrating the relative frequency of the six primary LTL temporal operators (G, F, X, U, R, W) within each of the 13 domains. This visualization highlights domain-specific tendencies in temporal patterns.

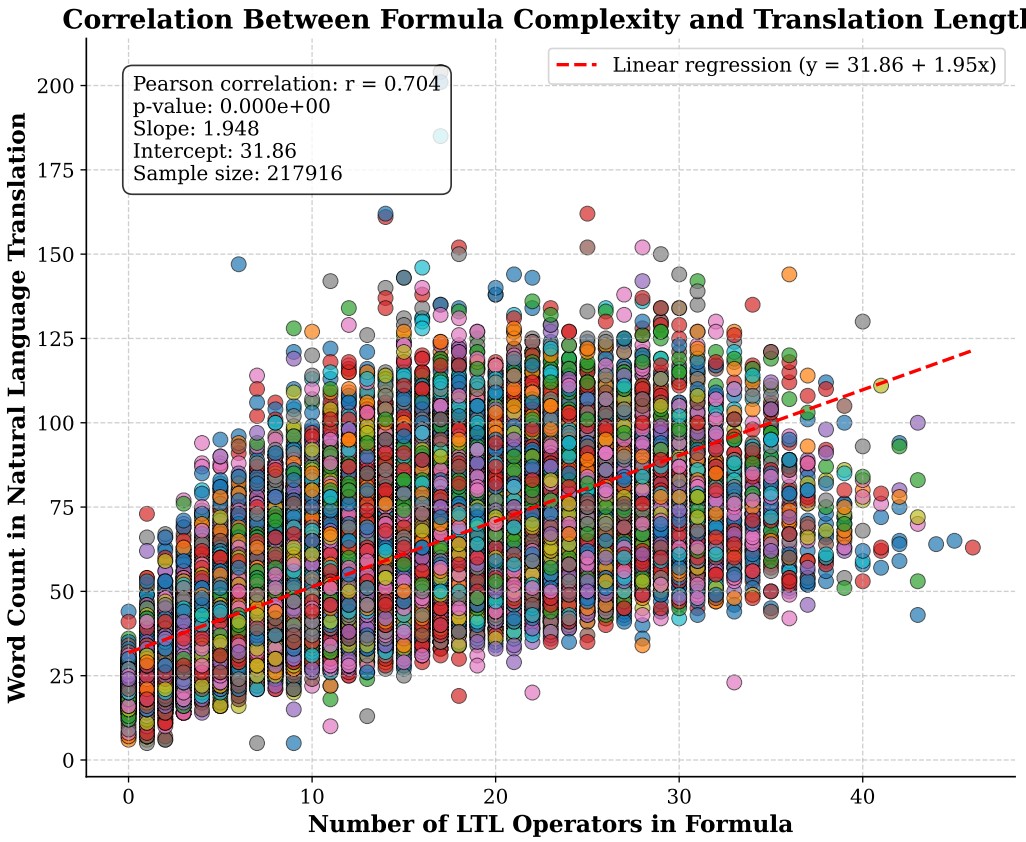

Figure 7: Scatter plot showing the correlation between LTL formula complexity (number of LTL operators) and the word count of the generated natural language translation. A linear regression line is overlaid. (Pearson correlation r=0.704, p < 0.001).

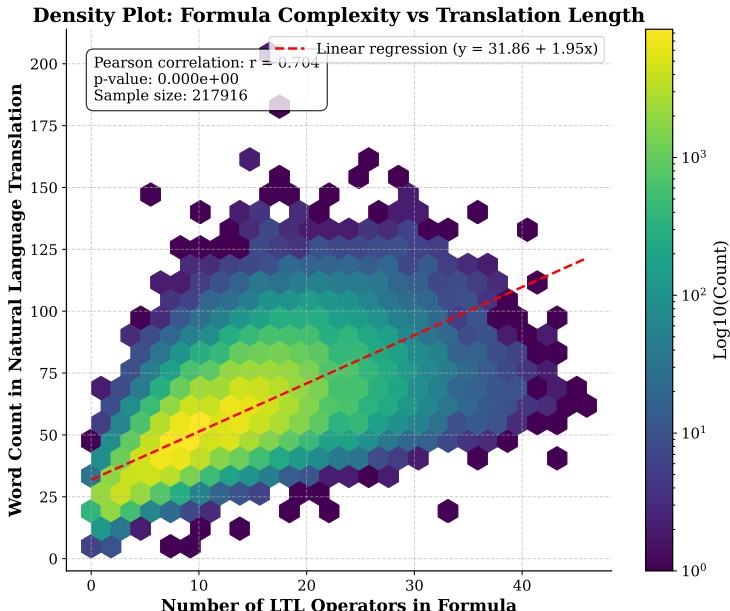

Figure 8: Density hexbin plot illustrating the relationship between LTL formula complexity (number of LTL operators) and the word count of the natural language translation. Darker regions indicate a higher concentration of data points. The linear regression line from Figure 7 is shown for reference.

## D.4 FULL DATA SCHEMA

The VERIFY dataset is released in standard CSV and Apache Parquet formats. Each record in the dataset represents a single LTL-ITL-NL triplet, along with its associated contextual information and metadata. The detailed schema is presented in Table 15, derived from the internal database structure.

Table 15: Full Data Schema for the VERIFY Dataset.

| Column Name | SQLite Type | CSV/Parquet Type | Constraints | Description |
|---|---|---|---|---|
| id | INTEGER | int | PRIMARY KEY (for triplet) | Unique identifier for the dataset record (triplet). |
| formula_id | INTEGER | int | NOT NULL; FK → conceptual formulas table | Identifier linking to a unique LTL formula structure (Spot-canonical). |
| itl_id | INTEGER | int | NOT NULL; FK → conceptual ITL table | Identifier linking to the unique canonical ITL structure (derived from formula_id). |
| domain | TEXT | string | NOT NULL | The application domain providing context (e.g., 'Aerospace'). |
| activity | TEXT | string | NOT NULL | Natural language definitions of the propositional variables used in the LTL formula, specific to the given domain. |
| ltl_formula | TEXT | string | NOT NULL | The formal LTL formula string (Spot-canonical representation using standard ASCII operators). |
| itl_representation | TEXT | string | NOT NULL | The corresponding canonical Intermediate Technical Language (ITL) string. |
| translation | TEXT | string | NOT NULL | The contextual Natural Language (NL) description corresponding to the LTL/ITL pair within the specified domain. |
| generation_time | REAL | float | | Time taken (in seconds) for the LLM to generate the 'activity' and 'translation' for this record. |
| timestamp | TEXT | string | | ISO 8601 timestamp indicating when the record (specifically the NL part) was generated. |

**Conceptual Data Relationships:**

- Each unique LTL formula (identified by formula_id after Spot canonicalization) has exactly one corresponding canonical ITL representation (identified by itl_id).

- A single LTL-canonical ITL pair (i.e., a unique `formula_id`) appears multiple times in the dataset, typically once for each of the 13 domains for which a natural language `translation` and `activity` definition were generated.

- The primary key `id` in the released files uniquely identifies each LTL-ITL-NL-Domain quadruplet.

**Example Raw Record:**

- **As a CSV data record (header shown first for clarity):**
```
id,formula_id,itl_id,domain,activity,ltl_formula,itl_representation,
translation,generation_time,timestamp

106230,22,229,"Aerospace",
"Consider an aircraft flight control system.  'p' indicates
the autopilot is engaged.  'q' indicates the aircraft is on
the correct flight path.  'r' indicates a critical system
failure.  's' indicates the flight director is providing
guidance.  't' indicates the aircraft has reached its
destination.  'u' indicates the aircraft is maintaining a
safe altitude.  'w' indicates the weather conditions are
within acceptable limits.",
"(p W q & w) W s U (!  (r -> r R u & q) U (!  X F p) | t) ->
s",
"if w and p weakly until q weakly until s until t or not
if r, then q and r releases u until not In the next state,
Eventually, p, then s",
"If acceptable weather and the autopilot being engaged
persist at least until the aircraft is on the correct flight
path, and this condition persists at least until the flight
director provides guidance, until the aircraft reaches
its destination or it is not the case that if there is
a critical system failure, the aircraft maintains a safe
altitude and the critical system failure continues to be
true at least until the aircraft is on the correct flight
path at least until it is not the case that in the next
state eventually the autopilot is engaged, then the flight
director provides guidance.",
1.54899549484253,"2025-04-30T12:51:08.943122"
```

- **As a JSON object representing one row (formatted for readability):**
```
{
  "id": 106230,
  "formula_id": 22,
  "itl_id": 229,
  "domain": "Aerospace",
  "activity": "Consider an aircraft flight control system. 'p' indicates \
the autopilot is engaged. 'q' indicates the aircraft is on the correct \
flight path. 'r' indicates a critical system failure. 's' indicates \
the flight director is providing guidance. 't' indicates the aircraft \
has reached its destination. 'u' indicates the aircraft is maintaining \
a safe altitude. 'w' indicates the weather conditions are within \
acceptable limits.",
  "ltl_formula": "(p W q & w) W s U (! (r -> r R u & q) U \
(! X F p) | t) -> s",
  "itl_representation": "if w and p weakly until q weakly until s until \
t or not if r, then q and r releases u until not In the next state, \
Eventually, p, then s",
  "translation": "If acceptable weather and the autopilot being engaged \
persist at least until the aircraft is on the correct flight path, and \
```

```
        this condition persists at least until the flight director provides \
        guidance, until the aircraft reaches its destination or it is not the \
        case that if there is a critical system failure, the aircraft \
        maintains a safe altitude and the critical system failure continues to \
        be true at least until the aircraft is on the correct flight path at \
        least until it is not the case that in the next state eventually the \
        autopilot is engaged, then the flight director provides guidance.",
      "generation_time": 1.54899549484253,
      "timestamp": "2025-04-30T12:51:08.943122"
    }
```

## E  IMPLEMENTATION DETAILS

This section details the methodologies and resources used for dataset generation, verification and the establishment of baseline experimental results.

### E.1  LTL FORMULA GENERATION AND VERIFICATION

**Generation:** Linear Temporal Logic (LTL) formulas were programmatically generated. The generation process recursively constructed formulas up to a maximum Abstract Syntax Tree (AST) depth of 25. This process utilized eight unique atomic propositions (denoted 'p' through 'w') and the standard LTL operators: Globally (G), Finally (F), Next (X), Until (U), Release (R), and Weak Until (W), along with boolean connectives ($\land, \lor, \neg, \rightarrow, \leftrightarrow$).

**Canonicalization and Uniqueness:** To ensure structural diversity and manage the formula space, generated LTL formulas underwent a rigorous canonicalization process. This involved conversion to Negation Normal Form (NNF), expansion of implications and equivalences, application of associative and distributive laws, and standardized sorting of operands for commutative operators. A unique hash was computed for each canonical structure to prevent duplicates in the master formula database, which was managed using SQLite.

**Formal Verification:** The semantic validity and non-triviality of all LTL formulas were formally verified using the Spot model checking library (version 2.11.6). A dedicated software component was designed to convert the LTL formulas from their generated format into Spot's required syntax. This component also managed the interaction with the Spot library for parsing, validation, and the retrieval of Spot's own canonical string representation for each formula. This ensured that all LTL formulas in the VERIFY dataset are well-formed and standardized according to a formal verification tool.

### E.2  INTERMEDIATE TECHNICAL LANGUAGE (ITL) GENERATION AND VERIFICATION

**AST-based Generation:** The canonical Intermediate Technical Language (ITL) representation for each verified LTL formula was generated deterministically. This process began by parsing the Spot-canonical LTL formula string into an internal AST representation, using Spot's parsing capabilities.

**Rule-based Transformation:** A rule-based transformation was then applied by traversing the LTL AST. For each LTL operator encountered in the AST, a corresponding human-readable template was selected from a predefined set of 12 core mapping rules (e.g., 'G $\phi$' maps to 'Always, $\phi$'; '$\phi$ U $\psi$' maps to '$\phi$ until $\psi$'). This mapping ensures that the resulting canonical ITL string directly preserves the structure of the source LTL formula while using more verbose, keyword-like operators.

**ITL Semantic Integrity:** To ensure the integrity of the ITL generation and its semantic equivalence to the source LTL, a verification step was implemented. This involved programmatically parsing the generated ITL text back into an LTL formula using a custom-designed recursive descent parser. This reconstructed LTL formula was then formally compared against the original, Spot-verified LTL formula. Equivalence was confirmed by ensuring that the Spot-canonical string representation of the reconstructed LTL formula matched that of the original Spot-canonical LTL formula. Spot's direct equivalence checking functions were also utilized during development for additional validation.

### E.3 NATURAL LANGUAGE (NL) GENERATION

**LLM and API Usage:** Contextual natural language descriptions (comprising domain-specific propositional variable 'activity' definitions and the 'translation' of the LTL/ITL logic) were generated using the DeepSeek-R1 model, specifically accessed via the 'deepseek-reasoner' API endpoint (model version available Q4 2024 - Q1 2025). This API was publicly available, subject to registration and usage quotas.

**Parallel Generation Orchestration:** To generate the extensive dataset, a parallel generation system was developed. This system orchestrated up to 500 concurrent Python 'asyncio' tasks distributed across multiple CPU nodes of an institutional high-performance computing (HPC) cluster. Each task handled an individual LTL/ITL pair for NL generation.

**Prompting Strategy:** For each LTL/ITL pair, a target domain was selected using a balanced sampling strategy designed to ensure roughly equal representation across the 13 diverse domains. The LLM was provided with the LTL formula, its ITL representation, and the selected domain. The prompt requested two specific outputs, encapsulated within XML-like tags:

```
Given the following formal specification:
LTL Formula: "{ltl_formula_string}"
Intermediate Technical Language (ITL): "{itl_representation_string}"
Domain: "{domain_name}"

Please perform two tasks:
1. Define plausible activities for the propositional variables used in the
LTL formula, relevant to the specified domain. These definitions should
make the LTL/ITL meaningful in that domain.
2. Translate the LTL/ITL formula into a clear, concise, and semantically
accurate natural language description. This description should incorporate
the domain context and the activities you define.

Format your response strictly as follows, ensuring the content within
the tags is on a single line if possible, or appropriately escaped if
multi-line:
<activity>p = [definition of p]; q = [definition of q]; ... </activity>
<translation>[Natural language translation of the LTL/ITL incorporating
the activities and domain context]</translation>
```

### E.4 LLM JUDGING FOR NL VALIDATION

**Validation Model:** A substantial portion (18%) of the generated NL translations underwent automated validation using a large language model to assess semantic correctness and quality. The model employed for this task was 'meta-llama/Meta-Llama-3-70B-Instruct'.

**Model Configuration:** The validation model was loaded using the Hugging Face 'transformers' library (version 4.49.0), with 8-bit quantization enabled via the 'bitsandbytes' library (version 0.45.5).

**Judging Prompt:** The LLM judge was provided with the LTL formula, its ITL representation, the generated NL translation, and the corresponding 'activity' string. It was tasked to output its assessment in a structured JSON format. The prompt template was as follows:

```
You are an expert in formal methods and natural language.
Your task is to evaluate the semantic correctness and quality
of a natural language (NL) translation with respect to a given
Linear Temporal Logic (LTL) formula and its Intermediate
Technical Language (ITL) representation.

LTL Formula: "{ltl_formula_string}"
ITL Representation: "{itl_representation_string}"
Generated NL Translation (incorporating domain context and
```

```
variable activities): "{nl_translation_string}"
Domain Context & Variable Activities (as generated for the NL
translation): "{activity_string_from_dataset}"

Please carefully assess the 'Generated NL Translation'.
Consider the following:
1.  Semantic Precision: Does the NL accurately convey the
    precise meaning of the LTL/ITL, especially the temporal
    relationships (e.g., always, eventually, until, next)?
2.  Contextual Appropriateness: Is the NL translation
    consistent with the provided 'Domain Context & Variable
    Activities'?
3.  Fluency & Clarity: Is the NL translation fluent,
    grammatically correct, and easily understandable?

Output your assessment *only* as a single JSON object with the
following keys:
- "is_correct": boolean (true if the NL translation is
  semantically correct with respect to LTL/ITL and
  contextually appropriate; false otherwise)
- "score": integer (an overall quality score from 0 to 10,
  where 10 is perfect)
- "issues": list of strings (a list of specific problems
  identified, e.g., "Misinterprets 'Until' operator",
  "NL is awkward". Empty list if no issues.)
- "reasoning": string (a brief textual explanation for your
  judgment and score.)
```

**Generation Parameters for Judge Output:** The generation of the JSON response by the LLM judge used the following parameters to ensure consistent and structured output: temperature = 0.1, top_p = 0.95, do_sample = True, and max_new_tokens = 512.

### E.5   BASELINE MODEL TRAINING

All baseline models were fine-tuned using a standardized methodology. Specific pre-trained checkpoints were sourced from the Hugging Face Hub. The Hugging Face 'transformers' library (version 4.49.0) and its 'Trainer' API were employed for the fine-tuning process. Data was tokenized using the respective model's default tokenizer, with input and output sequences padded or truncated to a maximum length of 512 tokens. LTL and ITL formulas were treated as regular text sequences for tokenization.

For each model and task, hyperparameters were optimized based on performance on the validation set, using the primary metric defined for that task (e.g., BERTScore F1 for LTL/ITL→NL, Semantic Equivalence for NL→LTL). The reported metrics in Tables 3, 4, and 5 of the main paper were calculated on the held-out test set using the best checkpoint identified during validation.

A representative configuration, exemplified by the **T5-large** model, is detailed below. Other models (T5-base, BART-base, BART-large, Llama-3-8B-Instruct, Mistral-7B-Instruct-v0.2, CodeLlama-7b-Instruct-hf, DeepSeek Coder-6.7b-instruct) followed an analogous fine-tuning procedure, adapting batch sizes and learning rates as appropriate for model size and stability.

**T5-large Example Configuration:**

- **Pre-trained Checkpoint:** 't5-large' (from Hugging Face Hub).

- **Task Input/Output Formatting:**
    - LTL/ITL → NL: Input: '"translate LTL to NL: domain: domain activity: activity ltl: LTL_formula"' (similarly for ITL). Output: '"NL_translation"'.
    - NL → LTL/ITL: Input: '"translate NL to LTL: domain: domain activity: activity nl: NL_translation"' (similarly for ITL). Output: '"LTL_formula"' or '"ITL_representation"'.

- LTL ↔ ITL: Input: '"translate LTL to ITL: ltl: LTL_formula"' (Output: '"ITL_representation"'), and vice-versa.

- **Training Hyperparameters:**
  - Learning Rate: Initial $1 \times 10^{-4}$, with a linear decay schedule.
  - Batch Size: 16 per device, with gradient accumulation steps of 4 (effective batch size of 64).
  - Training Epochs: 5.
  - Optimizer: AdamW ($\beta_1 = 0.9, \beta_2 = 0.999, \epsilon = 1 \times 10^{-8}$).
  - Weight Decay: 0.01.
  - Scheduler: Linear scheduler with warmup for the first 500 steps.
  - Gradient Clipping: Max norm of 1.0.

### E.6    SOFTWARE AND HARDWARE ENVIRONMENT

**Software Environment:** The primary development and execution environment utilized Python 3.10.16. Key libraries and their versions include:

- PyTorch (torch): 2.5.1+cu121
- Transformers (Hugging Face): 4.49.0
- Datasets (Hugging Face): 3.3.2
- Accelerate (Hugging Face): 1.4.0
- BitsandBytes: 0.45.5 (for 8-bit quantization)
- Spot (for LTL manipulation and verification): 2.11.6
- Pandas: 2.2.3
- NumPy: 1.26.4
- Scikit-learn: 1.6.1
- NLTK: 3.9.1 (for METEOR score)
- SQLite3 (Python standard library) for database management.

The CUDA version compatible with the PyTorch build and drivers was CUDA 12.1, with NVIDIA drivers version 550.x.

**Hardware Environment:** Dataset generation, initial processing, and LTL/ITL verification stages were primarily conducted on an institutional high-performance computing (HPC) cluster. These tasks utilized nodes equipped with dual AMD EPYC 9334 32-Core Processors. Natural language generation (orchestration of API calls) was also managed from these CPU-based HPC nodes. The LLM judging phase (using Llama 3) and all baseline model training and testing were performed on a separate institutional AI compute cluster. For LLM judging and training of larger baseline models (e.g., Llama-3-8B, Mistral-7B), nodes equipped with 8x NVIDIA H200 GPUs (141 GB VRAM per GPU) were utilized. Training of other baseline models (e.g., T5, BART variants) and final testing/evaluation across all models utilized nodes equipped with NVIDIA H100 GPUs (94 GB VRAM).

### E.7    COMPUTE RESOURCES

The development of the VERIFY dataset and the execution of baseline experiments required substantial computational resources.

- **LTL/ITL Database Generation & Verification:** Approximately 2,000 CPU core hours on AMD EPYC 9334 processors.
- **NL Generation (API Orchestration):** Approximately 72 wall-clock hours, heavily parallelized across multiple CPU nodes managing concurrent API calls. (The compute for the DeepSeek API itself is external).

- **LLM Judging (Llama 3):** Approximately 300 NVIDIA H200 GPU hours (for 18% of the dataset).

- **Baseline Model Training (average per model type):**
  - T5-large / BART-large type models: Approximately 24 hours on a 4x NVIDIA H100 GPU configuration.
  - T5-base / BART-base type models: Approximately 12 hours on a 4x NVIDIA H100 GPU configuration.
  - Llama 3 8B / Mistral 7B / CodeLlama 7B / DeepSeek Coder 6.7B type models: Approximately 36 hours on an 8x NVIDIA H200 GPU configuration.

