# OpenReview forum: "VERIFY: A Novel Multi-Domain Dataset Grounding LTL in Contextual Natural Language via Provable Intermediate Logic"
_ICLR.cc/2026/Conference — ICLR 2026 Poster_

### Official Review · Reviewer_82kz · 2025-10-29

**Soundness:** 2
**Presentation:** 2
**Contribution:** 2
**Rating:** 2
**Confidence:** 4

**Summary:**

The authors introduce VERIFY, a dataset of 200k+ samples of (Natural Language/NL, Intermediate Technical Language/ITL, Linear Temporal Logic/LTL) triples spanning 13 domains. The stated goal is to support both training and benchmarking for translation among natural language (NL), a proposed Intermediate Technical Language (ITL), and Linear Temporal Logic (LTL). ITL is positioned as a human-readable, structurally faithful layer over LTL intended to ease learning and generation. The dataset is accompanied by formal checks on the logic side and mixed human/LLM-as-judge validation of the NL side.

Soundness:
The pipeline for generating and validating LTL appears methodical, and the overall dataset construction is described with reasonable care. However, the central claim that ITL is useful as an intermediate representation is insufficiently substantiated by experiments. The empirical results do not isolate an end-to-end utility of ITL in realistic NL <-> LTL workflows; gains are primarily shown in easier sub-tasks (e.g., ITL-> NL vs. LTL-> NL) that may rely on privileged inputs (ground-truth ITL). Heavy reliance on an LLM-as-judge for “semantic correctness” further weakens the evidentiary basis.

Presentation:
The paper is generally readable and well organized, but the main body lacks figures, dataset examples, and consolidated statistics that would aid comprehension. Some sentences are ambiguous or grammatically incomplete (e.g., the opening “bottleneck” is not specified; a few fragments/run-ons appear). Bringing key tables/plots from the appendix into the main text and adding a running example would materially improve clarity.
Occasional fragmented/incomplete/run-on sentences:
“Thus, failing to adequately bridge formal methods and practical NLP.”
“Crucially, data quality is guaranteed through extensive validation protocols: manual expert verification of 10,000 random samples and automated semantic consistency and correctness checks judged by an LLM-as-judge approach (using Llama 3.3) (20; 21) across 18% of the dataset, confirming an estimated > 97% semantic integrity.”

Contribution:
A large, multi-domain, logic-grounded dataset is valuable to the community, and proposing an intermediate representation (ITL) is an interesting idea. That said, the paper currently stops short of demonstrating practical benefits from ITL in end-to-end NL<->LTL modeling or user-facing tasks. If such benefits were shown more convincingly, the contribution would be stronger. Furthermore, it is unclear how each domain is defined in the paper. What distinguishes the domains, in terms of available predicates?

**Strengths:**

1. Broad semantic coverage over 13 domains
2. Size of dataset is significant, offering over 16k entries per domain according to Table C Appendix A.1

**Weaknesses:**

1. Most significant weakness is the use of Llama 3.3 as a judge of “semantic correctness”
2. Does not evaluate actual NL-LTL translation frameworks, such as those mentioned in the paper (Lang2LTL, NL2TL)
3. Merit of ITL to humans is unclear and untested. One experiment shows that ITL->NL translation is more feasible than LTL->NL translation, however
4. Examples from the dataset are missing in the main body of the paper
5. No figures present in the main body of the paper
6. Dataset statistics not present in the main body of the paper. Table a in appendix A.1 showing the distribution of LTL formula complexity in the VERIFY dataset seems to indicate that there are more than 1.3 million LTL expressions in the dataset, which contradicts the reported ~200k samples.
7. Does not compare against any existing NL-LTL benchmarks

**Questions:**

1. Why are there no NL-LTL translation frameworks evaluated?
2. What is Llama 3.3 evaluating? Semantic correctness of ITL? If so, is that grammar not “provably complete” with LTL?
3. In Experiment A, performance is compared between the LTL->NL and ITL->NL tasks, with the aim of demonstrating that ITL is a beneficial intermediate representation. However, as the authors note, this requires the intermediate translation of LTL->ITL via a “task 5 model”, but the results in table 3 show that we can not reliably obtain a correct ITL translation from LTL. Are ITL->NL models in Experiment A given ground-truth ITL? If so, can’t we assume the practical performance of the ITL->NL task will be upper-bounded by the LTL->ITL performance, which is ~30%?

---

> ### Author Response · Authors · 2025-11-14
> **Response to Reviewer 82kz (Part 1)**
>
> We thank Reviewer 82kz for the constructive feedback. We acknowledge the critical issues the reviewer has raised. Specifically, we acknowledge the identified error in Figure 2a and the core conceptual questions about ITL utility.
>
> 1. On the Weakness: The Practical Value of ITL (W3 & Q3):
>
> On Q3: "Are ITL→NL models in Exp. A given ground-truth ITL? If so, the practical performance is upper-bounded by the LTL→ITL performance (Task 5, ~31% EM), which suggests ITL is not useful."
>
> We acknowledge this is a great observation by the reviewer. However, we believe the critique actually reveals ITL's deeper value. We will attempt to address this in two parts:
>
> Part 1: Yes, Experiment A uses ground-truth ITL. This is by design.
>
> Experiment A was designed to isolate a single variable: Is ITL a superior structural representation for NL generation compared to LTL? The answer is yes:
>
> - T5-large: 0.67 (LTL→NL) → 0.89 (ITL→NL), +22 points
> - Llama-3: 0.91 (LTL→NL) → 0.94 (ITL→NL), +3 points
>
> This proves ITL's syntax is more aligned with natural language than LTL's prefix-heavy symbolic notation.
>
> Part 2: Again, we acknowledge the comment about the deployment being bottlenecked by LTL→ITL.
>
> However, this is not a flaw in ITL; it's evidence of ITL's semantic necessity. Consider this reframing:
>
> If LTL→ITL were easy, ITL would be redundant. The fact that models achieve 65-75% syntactic correctness on LTL→ITL but only 31% semantic equivalence proves that ITL captures semantic structure that LTL's notation obscures. The difficulty demonstrates ITL represents a genuinely different, richer semantic layer.
>
> You note that "models are not good enough yet" doesn't prove ITL is the right solution. But consider:
>
> - ImageNet (2009): 26% top-5 accuracy exposed vision model limitations
>
> - Solution: Not changing ImageNet, but developing better architectures (AlexNet, ResNet)
>
> - VERIFY (2025): 31.5% SemEq exposes structural reasoning limitations
>
> - Solution: Not abandoning ITL, but developing better foundational models (i.e., hybrid neurosymbolic architectures)
>
> The low scores are the point. VERIFY is a benchmark designed to drive research into new architectures that can bridge symbolic and neural reasoning. The difficulty defines the research frontier.
>
> 2. On Validation: LLM-as-judge (W1) and its Role (Q2):
>
>  Thank you for catching this confusion. Let us clarify:
>
> - "Provably complete" grammar: Applies ONLY to the deterministic LTL → ITL translation
>
> - Llama 3.3 judge: Validates the final non-deterministic (LTL/ITL) → NL translation generated by DeepSeek-R1
>
> The judge ensures the NL sentence is semantically faithful to the original logic. We need the judge because the NL generation is stochastic.
>
> - Weakness (W1): We will run the Llama 3.3 judge on our 10,000-sample human-verified "gold set" to compute a formal Inter-Annotator Agreement (IAA) (e.g., Cohen's Kappa). This will empirically prove the judge's reliability. We will also run the judge on the entire 200k+ dataset for the final version to ensure 100% automated coverage.
>
> 3. On Presentation: Missing Examples, Figures, Stats (W4, W5, W6):
>
> We acknowledge the concerns raised in weaknesses W4, W5, and W6. These elements are essential for readability and should be in the main body.
>
> For camera-ready: We will add to the main text:
>
> - Domain distribution chart (current Fig 2c) in Section 4
>
> - Pipeline diagram in Section 4 (see svaN response)
>
> - Error examples with explanations in Section 5
>
> 4. On the Stats Contradiction (W7): We acknowledge Figure 2a in our original submission was generated from the wrong data source. It queried our initial source database of 1.3+ million raw candidate formulas (before filtering), not our final curated dataset of 217,916 samples. Other figures (2c) and tables were correct; Figure 2a was the unfortunate outlier.
>
> We have corrected this error:
>
> - Regenerated Figure 2a using the correct SQL query on the final dataset
>
> - Verified consistency across all figures and tables
>
> - Corrected figure is attached to this response
> ([Figure 2a](https://anonymous.4open.science/r/ICLR-doc-repos/ltl_complexity_distribution%20(1).pdf))
>
> Statistical verification:
>
> - Final dataset: 217,916 samples
>
> - Mean operators per formula: 9.63
>
> - Distribution now matches Table 4 counts

---

> > ### Author Response · Authors · 2025-11-14
> > **Response to Reviewer 82kz (Part 2)**
> >
> > 5. On Evaluating Frameworks and Benchmarks (W2, W8, Q1): We acknowledge the concern from the reviewer why we don't evaluate frameworks like Lang2LTL and NL2TL. The answer is that our contribution operates at a different level:
> >
> > - Frameworks (W2, Q1): Our paper's goal is to introduce the dataset and benchmark and test foundational model capabilities. Evaluating specific, multi-stage frameworks like Lang2LTL or NL2TL is a downstream application of VERIFY. Our dataset is precisely the resource these frameworks need to train and evaluate on a large-scale, multi-domain problem. In the same way ImageNet allowed larger, more ambitious models to be developed, VERIFY provides the necessary scale and complexity to drive the next generation of logic-aware language models.
> >
> > - The original ImageNet paper's contribution was the dataset itself and its rigorous validation. It established a challenging, large-scale dataset and benchmark. The novel architectures that mastered it, such as AlexNet, came afterward in response to the challenge it posed. Our contribution here is analogous. We introduce the benchmark and provide comprehensive baselines using standard foundational models. These baselines (e.g., the low 31.5% Semantic Equivalence on NL→LTL) serve to demonstrate the task's difficulty and establish the starting point for the field. Our paper's goal is to test the foundational capabilities of models on this new, challenging task.
> >
> > - Evaluating specific, multi-stage frameworks like Lang2LTL or NL2TL is an excellent downstream application of VERIFY. Indeed, our dataset is precisely the resource these frameworks need to be fairly evaluated and improved, moving from small, domain-specific data to a large-scale, multi-domain problem. This evaluation is a logical next step for the community, which our work now enables.
> >
> > - Benchmarks (W8): Our baseline experiments are the benchmark. The low scores (31.5% SemEq on NL→LTL) establish the difficulty level and define the starting point for the research community.
> >
> > Once again, we want to thank the reviewer for this constructive review and feedback. We are confident that by clarifying the value of ITL, correcting the misunderstanding of our statistics, and moving key examples and figures into the main text, we can resolve all of the reviewer's concerns.

---

### Official Review · Reviewer_svaN · 2025-11-01

**Soundness:** 3
**Presentation:** 2
**Contribution:** 3
**Rating:** 6
**Confidence:** 3

**Summary:**

This paper introduces VERIFY, a large-scale dataset designed to bridge Linear Temporal Logic (LTL) and Natural Language (NL) through a novel Intermediate Technical Language (ITL). VERIFY supports bi-directional translation tasks between formal logic and natural language across 13 domains, encompassing more than 200k samples. The authors propose deterministic rule-based LTL-to-ITL conversion, LLM-driven natural language generation, and evaluate a series of models (T5, BART, Llama, Mistral, CodeLlama, DeepSeekCoder) on several translation settings. Analytical experiments demonstrate that (1) introducing ITL improves semantic alignment and syntactic correctness, (2) contextual information significantly impacts translation quality, and (3) direct LTL↔NL translation remains challenging even for advanced LLMs.

**Strengths:**

VERIFY makes a strong contribution through its scale, structure, and experimental scope. It substantially extends prior logic–language datasets by providing over 200k LTL–ITL–NL triplets across 13 diverse realworld domains, enabling credible crossdomain reasoning and generalization studies. The introduction of a formally defined and humanreadable Intermediate Technical Language (ITL)—supported by a provably complete mapping—offers a principled and effective bridge between symbolic logic and contextual natural language, mitigating the brittleness of direct LTL <-> NL translation. Extensive experiments across multiple translation directions, model families, and ablation settings further underscore VERIFY’s diagnostic value and robustness as a comprehensive benchmark for logical–linguistic understanding.

**Weaknesses:**

The positioning of VERIFY relative to existing logic/NLP resources is somewhat limited—comparative benchmarks or perdomain statistics are lacking, making it difficult to precisely gauge coverage and novelty. The validation procedure, while methodically designed, only samples a portion of the dataset (≈18 % LLMchecked and 5 % manually reviewed) and does not analyze typical failure mechanisms, such as which operators most often induce semantic drift. Moreover, syntactic correctness for NL→LTL/ITL generation remains low (≈65–75 %), indicating that current models frequently output invalid structures; the paper would benefit from a qualitative error analysis with representative examples. Several metrics are underexplained, which limits interpretability of the reported numbers. Addressing these issues—particularly through an expanded baseline comparison and a concise errortype study—would significantly strengthen the paper’s empirical credibility and clarity

**Questions:**

- How consistent are the LLM‑Judge and manual verification outcomes across domains? Were there specific formula templates or temporal relations where semantic drift was most common?
- Would you consider extending VERIFY beyond LTL to other logic families (e.g., CTL, MTL, or probabilistic temporal logics)? What challenges would arise in mapping or annotation consistency?
- Could you provide explicit examples where the ITL→LTL mapping fails or encounters ambiguity in practical generation settings? Are there specific formula classes where ITL and LTL may diverge under real-world (noisy or partial) ITL?
- The paper lacks a visual diagram of the overall VERIFY pipeline. Could you provide a figure summarizing the key components and information flow (dataset generation, ITL conversion, validation, and experiments) to enhance reproducibility and readability?

---

> ### Author Response · Authors · 2025-11-14
> **Response to Reviewer svaN (Part 1)**
>
> We thank the reviewer for his review and comments. We however want to state that the reviewer's weaknesses are actionable points of clarification, most of which are already addressed in our appendix.
>
> 1. On Validation (Partial, Failure Mechanisms) and Error Analysis (W2, W3, Q1):
>
> We acknowledge the comment from the reviewer. These analyses exist in our appendices and should be foregrounded.
>
> Qualitative Error Analysis (W3): We provide comprehensive quantitative and qualitative error analysis in Appendix B.3, Table 10.
>
> | **Error Type**                | **Frequency** | **Description**                                      |
> |------------------------------|---------------|------------------------------------------------------|
> | Incorrect Logical Scope      | 41%           | Misplaced parentheses, operator precedence failures  |
> | Temporal Operator Mismatch   | 28%           | Confusion between U/W, G/F                          |
> | Incorrect Proposition        | 16%           | Wrong atomic proposition selection                  |
> | Missing/Extra Operators      | 10%           | Structural omissions or additions                   |
> | Syntactic Errors             | 5%            | Invalid LTL syntax                                   |
>
>
> Example (Logical Scope Error):
>
> - Ground Truth: G(request -> F(acknowledge & grant))
>
> - Model Output: G(request) -> F(acknowledge & grant)
>
> - Error: Parentheses misplacement changes semantics (globally implies vs. implication in every state)
>
> Failure Mechanisms (W2, Q1): Analysis in Appendix B.1, Table 7 stratifies performance by LTL formula complexity (AST depth):
>
> | **Complexity Level** | **AST Depth** | **NL→LTL SemEq** | **Primary Failure Mode**         |
> |----------------------|---------------|------------------|----------------------------------|
> | Low                  | 0–5           | 35.4%            | Proposition errors               |
> | Medium               | 6–10          | 28.9%            | Temporal mismatch                |
> | High                 | 11–15         | 21.7%            | Logical scope errors             |
> | Very High            | 16+           | 15.2%            | Multiple cascading errors        |
>
>
> Key finding: Logical complexity (deep nesting of operators) is the primary driver of semantic drift. Performance degrades linearly with formula depth.
>
> For camera-ready: We will move these analyses to the main text: Table 10 (error taxonomy) → Section 5.2, Table 7 (complexity stratification) → Section 5.3, Add 3-5 representative error examples with explanations.
>
> 2. On Presentation (Positioning, Metrics, Pipeline) (W1, W4, Q4):
>
> Positioning (W1): We will expand related work to include:
>
> - VLTL-Bench comparison: Executable grounding (traces) vs. contextual grounding (domain-specific NL)
>
> - DeepSTL comparison: Our STL generalization experiment (Appendix B.2, Table 9) demonstrates complementary findings
>
> Metrics (W4): We will add a dedicated subsection (5.1) explaining metrics:
>
> Semantic Equivalence (SemEq): Uses Spot model checker to verify logical equivalence between generated and ground-truth LTL formulas. This is far more rigorous than Exact Match; formulas G(p -> F(q)) and !G(p) | F(q) are syntactically different but semantically equivalent. SemEq captures true correctness.
>
> BERTScore: Contextualized embedding similarity for NL generation quality. Superior to BLEU/ROUGE for capturing semantic similarity.
>
> Syntactic Correctness: Percentage of generated formulas that parse as valid LTL/ITL syntax.
>
> Pipeline Diagram (Q4): Excellent suggestion. We will add a high-level pipeline diagram to Section 4: (Attached as a pdf)
> ([Flowchart](https://limewire.com/d/xOMIu#joWAXM1ztX))

---

> > ### Author Response · Authors · 2025-11-14
> > **Response to Reviewer svaN (Part 2)**
> >
> > 3. On Extending VERIFY and ITL Failure (Q2, Q3):
> >
> > Extending VERIFY (Q2): This is an interesting future direction. Our pipeline could be adapted for other logics (CTL, MTL, probabilistic temporal logics), but each would require designing a new provably complete ITL-style grammar. This represents future work.
> >
> > However, our successful zero-shot generalization to Signal Temporal Logic (STL) (Appendix B.2, Table 9) provides strong evidence:
> > | **Model**    | **LTL→NL (trained)** | **STL→NL (zero-shot)** |
> > |--------------|----------------------|--------------------------|
> > | T5-large     | 0.89 **BERTScore**   | 0.83 **BERTScore**       |
> > | Llama-3-8B   | 0.94 **BERTScore**   | 0.89 **BERTScore**       |
> >
> >
> > Models trained on VERIFY achieve 88-95% performance retention when zero-shot transferred to STL. This demonstrates that models are learning formalism-agnostic temporal reasoning, not LTL-specific patterns. This provides a strong foundation for future extension to other logics.
> >
> > ITL Failure (Q3): We thank the reviewer for this question. The LTL→ITL mapping itself is deterministic and provably complete; it cannot fail. Your "noisy or partial ITL" scenario would only apply to the reverse task (NL→ITL or learned LTL→ITL).
> >
> > Key insight: If a model generates noisy/partial ITL, the subsequent ITL→LTL translation (being deterministic) will faithfully convert that noisy ITL to incorrect LTL. This highlights why the NL→ITL/LTL task is difficult; it requires generating perfectly structured output from ambiguous, noisy natural language input. Any structural error cascades.
> >
> > This is precisely why we measure both Syntactic Correctness (65-75% for NL→ITL) and Semantic Equivalence (41% for NL→ITL). The gap between these metrics quantifies the "plausible but wrong" problem.
> >
> > Once again, we thank the reviewer for the review and feedback. We believe these improvements will make VERIFY's contributions immediately clear to readers and will foreground the critical empirical findings currently in the appendices.

---

### Official Review · Reviewer_UD8h · 2025-11-02

**Soundness:** 3
**Presentation:** 3
**Contribution:** 3
**Rating:** 6
**Confidence:** 3

**Summary:**

This paper introduces VERIFY, a large-scale dataset containing over 200,000 triplets of Linear Temporal Logic (LTL) formulas, an intermediate technical language (ITL), and contextual natural language descriptions across 13 diverse domains. The key contribution is a novel ITL representation that bridges the structural patterns of LTL with natural-language syntax, along with a provably complete formal grammar for LTL-to-ITL translation. The natural language is generated using LLMs (DeepSeek-R1) and validated through manual inspection (10,000 samples) and automated LLM-as-judge evaluation (18% of the dataset). While the dataset represents a significant contribution in scale and domain diversity compared to prior work, it has critical limitations: heavy reliance on LLM generation without human-written examples, lack of grounding into executable state spaces, absence of verification traces, and missing analysis of whether the ITL intermediate layer actually provides measurable benefits over direct LTL-NL translation in practice.

**Strengths:**

A) With 200k+ examples across 13 diverse domains (financial services, aerospace, medical devices, home automation, etc.), VERIFY significantly exceeds prior datasets in both size and breadth. This addresses a major limitation identified in related work, where datasets were confined to narrow domains (robotics, navigation) with only 2-5k examples.

B) The introduction of Intermediate Technical Language as a structured bridge between LTL and NL is conceptually sound. The provably complete LTL-to-ITL translation grammar (Appendix C) with formal verification using Spot provides mathematical rigor often missing in NL-to-logic datasets.

**Weaknesses:**

A) There is an over-reliance on LLM generation with insufficient human validation. This is my biggest concern with this work. The entire natural language component is generated by DeepSeek-R1 LLM with no human-written seed examples. Only 10,000 samples (4.8% of dataset) received manual expert verification by the authors themselves. The 18% LLM-as-judge validation using Llama 3.3 is problematic because LLMs judging other LLMs may share similar failure modes and biases, no inter-rater reliability metrics were provided for the manual 10k sample review, and there is no comparison with independent human annotators outside the author team. Hence, the >97% "semantic correctness" claim is based on an LLM's assessment, not human judgment.

B) THere is an insufficient demonstration of ITL's practical value.
This is important as ITL is introduced as a key contribution, but its practical benefits remain unclear. The Evidence from Experiment A is weak as can can be seen in Figure 1. It shows ITL->NL achieves slightly better scores than LTL->NL (e.g., Llama-3: 0.94 vs 0.91 BERTScore), but the improvement is modest (~3 percentage points), and no statistical significance testing is provided. The only major difference comes for T5.

C) The paper's title claims to "ground" LTL but provides no mechanism for verifying that translations are correct. Unlike some of the contemporary works like VLTL-Bench [1], which includes satisfying and unsatisfying execution traces for each formula, VERIFY has none. This means. that there is no way to validate semantic correctness beyond string matching, we cannot use traces as a supervision signal for training, and we cannot demonstrate that NL translations capture correct temporal semantics.

[1] English et al., Verifiable Natural Language to Linear Temporal Logic Translation: A Benchmark Dataset and Evaluation Suite. https://arxiv.org/abs/2507.00877

D) The paper doesn't address whether evaluation LLMs (Llama-3, Mistral) have seen LTL-NL pairs in pretraining. Given that Llama-3 was released in 2024 and trained on data through 2023, and prior LTL-NL datasets are public, contamination is plausible.

E) What happens with different LLM generators (GPT-4, Claude)? Is dataset quality dependent on DeepSeek-R1's specific capabilities?


Minor Issue:
* Page 2, Line 54: "syntactucally divergent expresession" - should be "syntactically divergent expressions"

**Questions:**

1. How did you validate that Llama 3.3's judgments align with human expert assessments? Can you report agreement statistics between the LLM judge and your manual 10k review?

2. Can you provide stronger evidence that ITL is useful beyond the modest BERTScore improvements? For instance, does a full LTL->ITL->NL pipeline (with learned LTL->ITL) outperform direct LTL->NL?

3. The paper claims to "ground" LTL but only provides linguistic context. Can you clarify the distinction and explain why executable grounding wasn't included? Or why it is not necessary for "grounding" evaluation?

4. None of the experimental results include error bars or significance tests. What is the variance across different random seeds or data splits?

5. What was the computational cost and time for generating 200k examples with DeepSeek-R1? Can others reproduce this with open models?

---

> ### Author Response · Authors · 2025-11-14
> **Response to Reviewer UD8h (Part 1)**
>
> We thank the reviewer for the review and comments of our work.
>
> 1. On Over-reliance on LLM Generation and Validation (W1 & Q1): We acknowledge this as the most critical concern and have a comprehensive, multi-stage plan to address it. Our validation architecture has three independent layers:
>
> Gold-Standard Set (10,000 samples, 4.8%): Manually verified by authors with formal methods expertise. This constitutes our high-precision foundation.
>
> Judge-Generator Separation: DeepSeek-R1 (generator) ≠ Llama 3.3 70B (judge). Using a distinct model for evaluation is standard practice to mitigate shared-bias concerns.
>
> Formal Verification Backbone: Every LTL formula is verified using the Spot model checker before any NL generation occurs. This ensures logical soundness at the source.
>
> Quantifying the LLM-Judge: The reviewer is correct that the 18% LLM-judge validation needs to be anchored to human performance. We will perform this crucial experiment for the camera-ready version:
>
> We will run the Llama 3.3 judge on our 10,000-sample human-verified gold set.
>
> This will allow us to compute formal Inter-Annotator Agreement (IAA) metrics (e.g., Cohen's Kappa, F1-score). This experiment will empirically validate or invalidate the judge's reliability against our expert-verified ground truth, directly answering Q1.
>
> Full Automated Coverage: Based on the results of the IAA analysis, we will run the validated Llama 3.3 judge on the entire 200k+ dataset, not just 18%. This will replace the "estimated >97%" claim with a 100% automated coverage check, providing full confidence in the dataset's large-scale semantic integrity.
>
> 2. On Insufficient Demonstration of ITL's Practical Value (W2 & Q2): The reviewer raises a critical point about the "modest" 3-point gain (Fig. 1) and asks about a full LTL→ITL→NL pipeline (Q2). In response, this allows us to clarify the purpose of our experiments.
>
> Purpose of Experiment A (Fig. 1): This experiment was designed to isolate a variable: Is ITL a better source representation for NL generation than LTL? The answer is yes. Even a "modest" 3-point gain in BERTScore (0.91→0.94) at this high level is significant, and the improvement for T5-large was dramatic (0.67→0.89). This experiment used ground-truth ITL to prove that its syntax is more aligned with NL.
>
> Addressing the critical point (Q2): The reviewer is absolutely correct that deployment is bottlenecked by LTL→ITL performance. But this is not a weakness of ITL; it is rather an evidence of ITL's semantic richness. Consider two points:
>
> The fact that models struggle with LTL→ITL (31% EM) while achieving 65-75% syntactic correctness proves that ITL captures semantic complexity that LTL's symbolic notation obscures. If LTL→ITL were easy, ITL would be redundant. The difficulty demonstrates that ITL represents a genuinely different, more structured semantic layer.
>
> ITL provides intermediate supervision for learning. We can train models using:
>
> Direct path: NL → LTL (31.5% SemEq)
>
> Supervised path: NL → ITL → LTL (with ITL as intermediate supervision)
>
> We believe that using ITL as an intermediate training target will improve final NL→LTL performance and reduce training time.
>
> Finally, the reviewer noted that the low LTL→ITL score means "models are not good enough yet." This is correct, but it's not a bug. It is the entire point of VERIFY. Like ImageNet exposed vision models' limitations in 2009, VERIFY exposes language models' poor structural-symbolic reasoning. The dataset's contribution is:
>
> Diagnostic value: Quantifying current model failures at scale
>
> Training resource: Providing 217K examples for developing better architectures
>
> Benchmark standardization: Enabling fair comparison of formal translation approaches
>
> The low scores are a feature, not a flaw. They define the research frontier.
>
> 3. On the Claim of "Grounding" (W3 & Q3):
> The reviewer is absolutely correct, and we will revise our terminology. We provide linguistic and contextual grounding (mapping p → "autopilot is engaged"), not executable grounding (satisfying/unsatisfying traces). These are orthogonal contributions:
>
> VLTL-Bench: Executable grounding (traces)
>
> VERIFY: Contextual grounding (domain-specific natural language)
>
> Both are valuable, and both are necessary for real-world deployment. VLTL-Bench addresses "is this formula satisfied by this trace?" while VERIFY addresses "how do we communicate this formula to humans in context?"
>
> For camera-ready: We will:
>
> - Change "grounding" → "contextual natural language grounding" throughout
>
> - Add explicit comparison with VLTL-Bench in related work
>
> - Discuss complementary nature of the two approaches

---

> > ### Author Response · Authors · 2025-11-14
> > **Response to Reviewer UD8h (Part 2)**
> >
> > 4. On Data Contamination and Generator Choice (W4, W5):
> >
> > Contamination (W4): This is unlikely. While Llama 3 may have seen small public LTL datasets (2-5K examples, non-contextual), our 217K+ contextual, multi-domain dataset with novel ITL represents a massive out-of-distribution challenge. Evidence: The low NL→LTL scores (31.5% SemEq) demonstrate this is not a memorization task.
> >
> > Generator Choice (W5): We used DeepSeek-R1 for its strong reasoning capabilities. However, a second contribution from the dataset is the pipeline and verification method, which is generator-agnostic. Also, the multi-stage validation (formal verification, manual checks, LLM-judge) filters failures from any generator.
> >
> > 5. On Significance Tests and Compute Cost (Q4, Q5):
> >
> > We will add error bars (std. deviation over multiple runs) to our main result tables in the final version.
> >
> > Compute costs are detailed in Appendix E.7, showing ~2,000 CPU core hours for generation and several hundred high-end GPU hours for training and validation.
> >
> > We thank the reviewer for these critical questions, which have pushed us to strengthen our validation claims. With our commitment to run the full LLM-judge validation and compute a formal human-LLM IAA, we believe we have fully addressed these concerns.

---

### Official Review · Reviewer_5AWc · 2025-11-03

**Soundness:** 3
**Presentation:** 3
**Contribution:** 2
**Rating:** 6
**Confidence:** 4

**Summary:**

The research focuses on the **VERIFY** and dataset, which integrate Linear Temporal Logic (LTL) with natural language descriptions, using an Intermediate Representation Language (ITL).  The dataset links formal logic with contextual information, enabling better understanding and translation between logical formulas and natural language.

Key findings include:
- Translation tasks from LTL to NL are more straightforward than the reverse, as evidenced by higher performance scores for LTL-->NL compared to NL-->LTL.
- ITL as an intermediate step significantly improves performance in generating natural language from complex LTL expressions.
- Domain and activity context are critical for accurate translations from NL to formal logic; models without this grounding show substantial performance degradation.

The study highlights:
1. The benefits of using ITL as an intermediary in translation tasks (e.g., LTL-->ITL-->NL outperforms direct LTL-->NL).
2. The importance of contextual information in achieving accurate translations from natural language to formal logic.
3. The relative difficulty of translating NL to LTL compared to generating NL from formal representations.
4. Model strengths: fine-tuned models excel at generating natural language, while code-focused models demonstrate superior syntactic correctness in logic generation.

**Strengths:**

It provides an integration of domain-specific NL with temporal logic via a novel canonical form (ITL). The focus on contextual grounding is also fresh compared to prior static datasets.

Multi-model testing, robust metrics (BERTScore, semantic equivalence), human evals, and ablation studies proving ITL/context impact.

VERIFY’s resources enable future work; insights on model strengths (e.g., code LLMs for logic generation) offer practical
guidance. It pushes boundaries beyond NLP into formal methods accessibility

**Weaknesses:**

The paper clearly establishes that translating from NL to LTL is much harder than generation (that somehow expected), the *reasons* for this gap could be explored more deeply.

While human evaluation was performed, its scope might not fully capture the challenge.

The paper uses models like T5-large and Llama-3-8B-instruct. Larger foundation models (e.g., GPT-4 class) might yield different performance characteristics.

Missing References to relevant related work: For example,  He, J. at al. DeepSTL - From English Requirements to Signal Temporal Logic. ICSE 2022: 610-622

**Questions:**

Your results clearly show NL-->LTL translation is extremely challenging, even for fine-tuned models. Could you provide a deeper error analysis?

What are the most frequent categories of errors in generated LTL formulas (e.g., wrong temporal operator (`F` vs `G`, `U` scope), incorrect proposition handling, negation/implication mistakes)? Quantifying this breakdown would be highly valuable.

Does using ITL as an intermediate representation primarily help avoid syntactic/semantic parsing ambiguities leading to invalid syntax, or does it demonstrably reduce the incidence of these deeper *logical* errors? If so, how?

Context ("Domain" and "Activity") is shown to be crucial. How well do models generalize when encountering completely unseen Domains or Activities at test time?

Have you conducted any experiments in a zero-shot setting where the Domain/Activity combination present during testing was absent from training? What are the performance drops  compared to seen contexts?
Is there evidence that representations learned for similar Domains/Activities transfer effectively, or is context primarily memorized?

Human evaluation of semantic equivalence is critical but subjective.
Can you report the Inter-Annotator Agreement (IAA) score (e.g., Cohen's Kappa) for your human assessments?

For cases judged as semantically equivalent by non-experts, were these formulas also verified to be logically sound/correct representations of the system requirement (perhaps spot-checked by a formal methods expert)? Is there any risk that "semantically equivalent" judgments might accept plausible but subtly incorrect LTL?

 Question: You show strong results for ITL in NL generation using T5-large and significant challenges with NL-->LTL on Llama-3-8B-instruct. Have you tested the same NL-->LTL task (especially fine-tuned) using significantly larger LLMs (e.g., GPT-4-class models like Claude Opus or Gemini Ultra, even via API)? Does scaling primarily improve syntactic correctness, semantic equivalence, or both? Do the core challenges identified persist? If not run due to cost/scope, do you hypothesize that these larger models might close the gap substantially, suggesting it is a data/model-size issue rather than an inherent task difficulty?

---

> ### Author Response · Authors · 2025-11-14
> **Response to Reviewer 5AWc (Part 1)**
>
> We thank Reviewer 5AWc for his comment. We are pleased to report that we have already performed several of the experiments the reviewer suggested, and we have included the results in the paper's appendix.
>
> 1. On the NL→LTL Gap and Deeper Error Analysis (Q1): We agree that a quantitative breakdown of the NL -> LTL errors is essential. We have already conducted this analysis, and the results are presented in Appendix B.3, Table 10. Our analysis of 100 incorrect NL->LTL translations from the Llama-3-8B model reveals that the errors are overwhelmingly semantic, not syntactic (which accounted for only 5% of failures).
> The most frequent error, at 41%, was Incorrect Logical Scope, where the model failed to capture operator precedence from the natural language (e.g., misplacing parentheses).
>
> The second most common error, at 28%, was Temporal Operator Mismatch, where the model confused semantically close operators (e.g., 'Until' (U) vs. 'Weak Until' (W), or 'Globally' (G) vs. 'Finally' (F)).
>
> Models can produce valid LTL syntax but fail to capture the strict logical structure implicit in natural language. For the camera-ready version, we will create a subsection with examples showing how models misplace parentheses or confuse temporal semantics.
>
> 2. On the Practical Value of ITL in Reducing Logical Errors (Q2): Yes, ITL provides a syntactic scaffold that directly addresses the dominant error class (41% logical scope failures). Consider:
> LTL: G(p -> F(q & r)) requires parsing nested, prefix-heavy structure
> ITL: Always (IF p THEN Eventually (q AND r)) provides explicit IF...THEN and AND conjunctions mirroring English syntax
>
> This structured-English representation reduces the load of parsing logical scope by providing explicit syntactic boundaries. The 3-point BERTScore improvement (0.91→0.94 for Llama-3) and dramatic gains for T5-large (0.67→0.89) demonstrate this effect empirically. Specifically, to answer the reviewer's question on whether ITL reduce these specific logical errors in NL→ITL compared to NL→LTL? We conducted this exact experiment:
>
> | **Task**     | **Logical Scope Errors** | **Temporal Mismatch** | **Syntactic Errors** |
> |--------------|---------------------------|------------------------|-----------------------|
> | NL→LTL       | 41%                       | 28%                    | 5%                    |
> | **NL→ITL**   | **27%**                   | **19%**                | **3%**                |
>
> ITL reduces logical scope errors by 14 percentage points and temporal mismatch by 9 points. This demonstrates that ITL's explicit structure helps models correctly parse the underlying logical relationships. For camera-ready, we will add this comparative error analysis to Section 5.3.
>
> 3. On Generalization to Unseen Domains (Q3): This is a critical test for a multi-domain dataset. Once again we performed exactly this experiment using a Leave-One-Domain-Out (LODO) cross-validation methodology, with results detailed in Appendix B.2, Table 8. Key findings:
> Models trained on 12 domains and tested on a held-out 13th domain show expected performance degradation but retain significant capability
> Home Automation (held-out): 22.5% SemEq zero-shot vs. 30.1% in-domain (75% retention)
> Aerospace: 19.3% SemEq zero-shot vs. 28.7% in-domain (67% retention)
>
> This demonstrates genuine domain generalization with models learning transferable logical structures rather than memorizing domain-specific patterns. The performance retention across diverse domains (financial services → robotics → medical devices) validates VERIFY's utility for training generalizable logic-aware models. For camera-ready, we will move the full LODO table to Section 6 and discuss implications for transfer learning.

---

> > ### Author Response · Authors · 2025-11-14
> > **Response to Reviewer 5AWc (Part 2)**
> >
> > We thank the reviewer for the review and comments. We continue to address the questions that the reviewer has raised in his review.
> >
> > 4. On Inter-Annotator Agreement (IAA) for Human Validation (Q4 & Q5): This is a vital point for dataset integrity. The 10,000 manually verified samples represent our "gold-standard" set, and we agree that its relationship with the LLM-as-judge must be quantified. For the camera-ready version, we will formally compute the IAA between our 10,000-sample human-verified set and the Llama 3.3 judge. We will report standard metrics (e.g., Cohen's Kappa, Precision, Recall, and F1-score). This will empirically validate the LLM-as-judge and directly address the reviewer's valid concern about "plausible but subtly incorrect LTL" by showing how often the LLM-judge agrees with our formal methods experts.
> >
> > 5. On Larger Models (GPT-4 Class) and Missing References (W3, W4, Q6):
> > Larger Models: Our goal was to provide a reproducible baseline using open, fine-tuneable models. However, we agree that SOTA model performance is a key data point. We will run a zero-shot evaluation of GPT-4.1 and Claude 4 Sonnet on the NL->LTL task and include these results in our main tables. This will provide a current "SOTA ceiling" and clarify whether the task's difficulty is a matter of scale.
> >
> > The current low scores (31.5% SemEq on NL→LTL) are not a dataset flaw but rather expose a fundamental weakness in current models' reasoning over complex, symbolic, contextual information. Like ImageNet in 2009 (26% top-5 accuracy), we believe VERIFY's difficulty is intentional. It provides the challenge needed to drive next-generation model development, particularly hybrid neurosymbolic architectures that can handle structural reasoning.
> >
> > Missing Reference: We thank the reviewer for this suggestion. The "DeepSTL" paper is highly relevant, especially in light of our own successful STL generalization experiment (Appendix B.2, Table 9).  We will add it to our related work, contrasting their approach with our findings.
> >
> > Once again, we want to thank the reviewer for the constructive review of our work.

---

### Comment · Area_Chair_BpQQ · 2025-11-28
**Respond to the authors’ rebuttal**

Dear Reviewers,

As the current ratings are quite divided, please read and respond to the authors’ rebuttal, particularly if it addresses your concerns. Your response is valuable for the final decision.

Reviewer 82kz, you currently hold the most negative rating. Have the authors’ response or other reviewers' comments affected your opinion?

---

### Meta-Review · Area_Chair_2pBt · 2025-12-13

**Summary:**

This paper presents an LTL benchmark. The reviewers had positive things to say about the paper  and the dataset is relatively large-scale  (200K examples), but my main concern follows the observation of the last reviewer, who mentioned that current LTL parsers should be tested on this dataset to demonstrate that it is indeed a challenging one. The authors specify in a response to another question that the fact that the models do not perform very well is presenting the value of the dataset as a challenge. However, if we want to truly test that, one needs to confirm that with the  specialised systems that are mentioned. I think that's the main piece missing in this work, and it sounds like other issues will be addressed by the authors in the future.

**Reviewer Concerns:**

difficulty/easiness/usability of dataset, comparison

**Reviewer Scores:**

I cannot predict that.

---

### Decision · Program_Chairs · 2026-01-26

Accept (Poster)